# The Nature of Temporal Difference Errors in Multi-step Distributional Reinforcement Learning

**Yunhao Tang**
DeepMind
robintyh@deepmind.com

**Mark Rowland**
DeepMind
markrowland@deepmind.com

**Rémi Munos**
DeepMind
munos@deepmind.com

**Bernardo Ávila Pires**
DeepMind
bavilapires@deepmind.com

**Will Dabney**
DeepMind
wdabney@deepmind.com

**Marc G. Bellemare**
Google Brain
bellemare@google.com

## Abstract

We study the multi-step off-policy learning approach to distributional RL. Despite the apparent similarity between value-based RL and distributional RL, our study reveals intriguing and fundamental differences between the two cases in the multi-step setting. We identify a novel notion of path-dependent distributional TD error, which is indispensable for principled multi-step distributional RL. The distinction from the value-based case bears important implications on concepts such as backward-view algorithms. Our work provides the first theoretical guarantees on multi-step off-policy distributional RL algorithms, including results that apply to the small number of existing approaches to multi-step distributional RL. In addition, we derive a novel algorithm, Quantile Regression-Retrace, which leads to a deep RL agent QR-DQN-Retrace that shows empirical improvements over QR-DQN on the Atari-57 benchmark. Collectively, we shed light on how unique challenges in multi-step distributional RL can be addressed both in theory and practice.

## 1 Introduction

The return $\sum_{t=0}^{\infty} \gamma^t R_t$ is a fundamental concept in reinforcement learning (RL). In general, the return is a random variable, whose distribution captures important information such as the stochasticity in future events. While the classic view of value-based RL typically focuses on the expected return [1–3], learning the full return distribution is of both theoretical and practical importance [4–10].

To design efficient algorithms for learning return distributions, a natural idea is to construct distributional equivalents of existing multi-step off-policy value-based algorithms. In value-based RL, multi-step learning tends to propagate useful information more efficiently and off-policy learning is ubiquitous in modern RL systems. Meanwhile, the return distribution shares inherent commonalities with the expected return, thanks to the close connection between the distributional Bellman equation [4–6, 10] and the celebrated value-based Bellman equation [2]. The Bellman equation is foundational to value-based RL algorithms, including many multi-step off-policy methods [11–14]. Due to the apparent similarity between distributional and value-based Bellman equations, should we expect key value-based concepts and algorithms to seamlessly transfer to distributional learning?

Our study indicates that the answer is no. There are critical differences between distributional and value-based RL, which requires a distinct treatment of multi-step learning. Indeed, thanks to the focus on expected returns, the value-based setup offers many unique conceptual and computational simplifications in algorithmic design. However, we find that such simplifications do not hold for distributional learning. Multi-step distributional RL requires a deeper look at the connections between fundamental concepts such as $n$-step returns, TD errors and importance weights for off-policy learning. To this end, we make the following conceptual, theoretical and algorithmic contributions:

36th Conference on Neural Information Processing Systems (NeurIPS 2022).

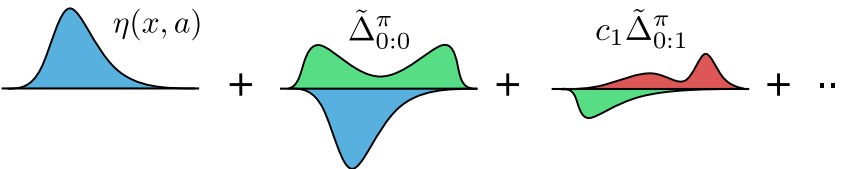

Figure 1: Illustration of a multi-step distributional RL target, constructed as a sum of the initial distribution (left) and weighted distributional TD errors $\widetilde{\Delta}_{0:0}^\pi, c_1\widetilde{\Delta}_{0:1}^\pi, \ldots$ across multiple time steps (middle and right); see Section 3 for further details and notation. In general, distributional TD errors are signed measures, as reflected by the downwards probability mass; they are also scaled by trace coefficients $c_1$ to correct for off-policy discrepancies between target and behavior policy.

**Distributional TD error.** We demonstrate the emergence of a novel notion of path-dependent distributional TD error (Section 4). Intriguingly, as the name suggests, path-dependent distributional TD errors are *path-dependent*, i.e., distributional TD errors at time $t$ depend on the sequence of immediate rewards $(R_s)_{s=0}^{t-1}$. This differs from value-based TD errors, which are path-independent. We will show that the path-dependency property is not an artifact, but rather a fundamental property of distributional learning. We show numerically that naively constructing certain path-independent distributional TD errors does not produce convergent algorithms. The path-dependency property also has conceptual and computational impacts on forward-view estimates and backward-view algorithms.

**Theory of multi-step distributional RL.** We derive distributional Retrace, a novel and generic multi-step off-policy operator for distributional learning. We prove that distributional Retrace is contractive and has the target return distribution as its fixed point. Distributional Retrace interpolates between the one-step distributional Bellman operator [6] and Monte-Carlo (MC) estimation with importance weighting [15], trading-off the strengths from the two extremes.

**Approximate multi-step distributional RL.** Finally, we derive Quantile Regression-Retrace, a novel algorithm combining distributional Retrace with quantile representations of distributions [16] (Section 5). One major technical challenge is to define the quantile regression (QR) loss against signed measures, which are unavoidable in sample-based settings. We bypass the issue of ill-defined QR loss and derive unbiased stochastic estimates to the QR loss gradient. This leads up to QR-DQN-Retrace, a deep RL agent with performance improvements over QR-DQN on Atari-57 games.

In Figure 1, we illustrate how the back-up target is computed for multi-step distributional RL. In summary, we take our findings to demonstrate how the set of unique challenges presented by multi-step distributional RL can be addressed both theoretically and empirically. Our study also opens up many exciting research pathways in this domain, paving the way for future investigations.

## 2 Background

Consider a Markov decision process (MDP) represented as the tuple $(\mathcal{X}, \mathcal{A}, P_R, P, \gamma)$ where $\mathcal{X}$ is the state space, $\mathcal{A}$ the action space, $P_R : \mathcal{X} \times \mathcal{A} \to \mathscr{P}(\mathscr{R})$ the reward kernel (with $\mathscr{R}$ a finite set of possible rewards), $P : \mathcal{X} \times \mathcal{A} \to \mathscr{P}(\mathcal{X})$ the transition kernel and $\gamma \in [0, 1)$ the discount factor. In general, we use $\mathscr{P}(A)$ denote a distribution over set $A$. We assume the reward to take a finite set of values mainly because it is notationally simpler to present results; it is straightforward to extend our results to the general case. Let $\pi : \mathcal{X} \to \mathscr{P}(\mathcal{A})$ be a fixed policy. We use $(X_t, A_t, R_t)_{t=0}^\infty \sim \pi$ to denote a random trajectory sampled from $\pi$, such that $A_t \sim \pi(\cdot|X_t), R_t \sim P_R(\cdot|X_t, A_t), X_{t+1} \sim P(\cdot|X_t, A_t)$. Define $G^\pi(x, a) := \sum_{t=0}^\infty \gamma^t R_t$ as the random return, obtained by following $\pi$ starting from $(x, a)$. The Q-function $Q^\pi(x, a) := \mathbb{E}[G^\pi(x, a)]$ is defined as the expected return under policy $\pi$. For convenience, we also adopt the vector notation $Q \in \mathbb{R}^{\mathcal{X} \times \mathcal{A}}$. Define the one-step value-based Bellman operator $T^\pi : \mathbb{R}^{\mathcal{X} \times \mathcal{A}} \to \mathbb{R}^{\mathcal{X} \times \mathcal{A}}$ such that $T^\pi Q(x, a) := \mathbb{E}[R_0 + \gamma Q(X_1, A_1^\pi)|X_0 = x, A_0 = a]$ where $Q(X_t, A_t^\pi) := \sum_a \pi(a|X_t)Q(X_t, a)$. The Q-function $Q^\pi$ satisfies $Q^\pi = T^\pi Q^\pi$ and is also the unique fixed point of $T^\pi$.

## 2.1 Distributional reinforcement learning

In general, the return $G^\pi(x, a)$ is a random variable and we define its distribution as $\eta^\pi(x, a) :=$ $\text{Law}_\pi(G^\pi(x, a))$. The return distribution satisfies the distributional Bellman equation [4–6, 17, 10],

$$\eta^\pi(x, a) = \mathbb{E}_\pi\left[(\mathsf{b}_{R_0, \gamma})_\# \, \eta^\pi(X_1, A_1^\pi) \mid X_0 = x, A_0 = a\right], \tag{1}$$

where $(\mathsf{b}_{r,\gamma})_\# : \mathscr{P}(\mathbb{R}) \to \mathscr{P}(\mathbb{R})$ is the pushforward operation defined through the function $\mathsf{b}_{r,\gamma}(z) = r + \gamma z$ [17]. For convenience, we adopt the notation $\eta^\pi(X_t, A_t^\pi) := \sum_a \pi(a|X_t)\eta^\pi(X_t, a)$. Throughout the paper, we focus on the space of distributions with bounded support $\mathscr{P}_\infty(\mathbb{R})$. Let $\eta \in \mathscr{P}_\infty(\mathbb{R})^{\mathcal{X} \times \mathcal{A}}$ be any distribution vector, we define the *distributional Bellman operator* $\mathcal{T}^\pi : \mathscr{P}_\infty(\mathbb{R})^{\mathcal{X} \times \mathcal{A}} \to \mathscr{P}_\infty(\mathbb{R})^{\mathcal{X} \times \mathcal{A}}$ as follows [17, 10],

$$\mathcal{T}^\pi\eta(x, a) := \mathbb{E}\left[(\mathsf{b}_{R_0, \gamma})_\# \eta(X_1, A_1^\pi) \mid X_0 = x, A_0 = a\right]. \tag{2}$$

Let $\eta^\pi$ be the collection of return distributions under $\pi$; the distributional Bellman equation can then be rewritten as $\eta^\pi = \mathcal{T}^\pi\eta^\pi$. The distributional Bellman operator $\mathcal{T}^\pi$ is $\gamma$-contractive under the supremum $p$-Wasserstein distance [16, 10], so that $\eta^\pi$ is the unique fixed point of $\mathcal{T}^\pi$. See Appendix B for details of the distance metrics.

## 2.2 Multi-step off-policy value-based learning

We provide a brief background on the value-based multi-step off-policy setting as a reference for the distributional case discussed below. In off-policy learning, the data is generated under a behavior policy $\mu$, which potentially differs from target policy $\pi$. The aim is to evaluate the target Q-function $Q^\pi$. As a standard assumption, we require $\text{supp}(\pi(\cdot|x)) \subseteq \text{supp}(\mu(\cdot|x)), \forall x \in \mathcal{X}$. Let $\rho_t := \pi(A_t|X_t)/\mu(A_t|X_t)$ be the step-wise importance sampling (IS) ratio at time step $t$. Step-wise IS ratios are critical in correcting for the off-policy discrepancy between $\pi$ and $\mu$.

Let $c_t \in [0, \rho_t]$ be a time-dependent trace coefficient. We denote $c_{1:t} = c_1 \cdots c_t$ and define $c_{1:0} = 1$ by convention. Consider a generic form of the return-based off-policy operator $R^{\pi,\mu}$ as in [13],

$$R^{\pi,\mu}Q(x, a) := Q(x, a) + \mathbb{E}_\mu\left[\sum_{t=0}^\infty c_{1:t}\gamma^t \underbrace{\left(R_t + \gamma Q\left(X_{t+1}, A_{t+1}^\pi\right) - Q(X_t, A_t)\right)}_{\delta_t^\pi = \text{value-based TD error}}\right], \tag{3}$$

In the above and below, we omit the notation conditioning on $X_0 = x, A_0 = a$ for conciseness. The general form of $R^{\pi,\mu}$ encompasses many important special cases: when on-policy and $c_t = \lambda$, it recovers the Q-function variant of TD($\lambda$) [2, 12]; when $c_t = \lambda \min(\bar{c}, \rho_t)$, it recovers a specific form of Retrace [13]; when $c_t = \rho_t$, it recovers the importance sampling (IS) operator. The back-up target is computed as a mixture over TD errors $\delta_t^\pi$, each calculated from the one-step transition data. We also define the *discounted TD error* $\widetilde{\delta}_t^\pi = \gamma^t \delta_t^\pi$, which can be interpreted as the difference between $n$-step returns from two time steps $t$ and $t + 1$, as we discuss in Section 4. As we will detail, the property of $\widetilde{\delta}_t^\pi$ marks a significant difference from the distributional RL setting.

By design, $R^{\pi,\mu}$ has $Q^\pi$ as the unique fixed point. Multi-step updates make use of rewards from multiple time steps, propagating learning signals more efficiently. This is reflected by the fact that $R^{\pi,\mu}$ is $\beta$-contractive with $\beta \in [0, \gamma]$ [13] and often contracts to $Q^\pi$ faster than the one-step Bellman operator $T^\pi$. Our goal is to design distributional equivalents of multi-step off-policy operators, which can lead to concrete algorithms with sample-based learning.

## 3 Multi-step off-policy distributional reinforcement learning

We now present the core theoretical results relating to multi-step distributional operators. In general, the aim is to evaluate the target distribution $\eta^\pi$ with access to off-policy data generated under $\mu$.

Below, we use $G_{t':t} = \sum_{s=t'}^t \gamma^{s-t'} R_s$ to denote the partial sum of discounted rewards between two time steps $t' \leq t$. We define the generic form of multi-step off-policy distributional operator $\mathcal{R}^{\pi,\mu}$ such that for any $\eta \in \mathscr{P}_\infty(\mathbb{R})^{\mathcal{X} \times \mathcal{A}}$, its back-up target $\mathcal{R}^{\pi,\mu}\eta(x, a)$ is computed as

$$\eta(x, a) + \mathbb{E}_\mu\left[\sum_{t=0}^\infty c_{1:t} \cdot \underbrace{\left((\mathsf{b}_{G_{0:t}, \gamma^{t+1}})_\# \eta\left(X_{t+1}, A_{t+1}^\pi\right) - (\mathsf{b}_{G_{0:t-1}, \gamma^t})_\# \eta(X_t, A_t)\right)}_{\widetilde{\Delta}_{0:t}^\pi = \text{Multi-step Distributional TD error}}\right]. \tag{4}$$

As an effort to simplify the naming, we call $\mathcal{R}^{\pi,\mu}$ the *distributional Retrace* operator. Distributional Retrace only requires $c_t \in [0, \rho_t]$ and represents a large family of distributional operators. Throughout, we will heavily adopt the pushforward notations. This is mainly because instead of directly working with the random variable $G^\pi$, we find it much more convenient to express various important multi-step operations with pushfoward notations.

The back-up target $\mathcal{R}^{\pi,\mu}\eta(x, a)$ is written as a weighted sum of the path-dependent distributional TD errors $\widetilde{\Delta}_{0:t}^\pi$, which we extensively discuss in Section 4. Though the form of $\mathcal{R}^{\pi,\mu}$ seems to bear certain similarities to the value-based operator in Equation (3), the critical differences lie in subtle definitions of the distributional TD errors $\widetilde{\Delta}_{0:t}^\pi$ and where to place the traces $c_{1:t}$ for off-policy corrections. We resume to unpack the insights entailed by the design of the operator in Section 4.

Below, we first present theoretical properties of the distributional Retrace operator. We start with a key property which underlies many ensuing theoretical results. Given a fixed $n$-step reward sequence $r_{0:n-1}$ and a fixed state-action pair $(x, a) \in \mathcal{X} \times \mathcal{A}$, we call pushfoward distributions of the form $\left( b_{\sum_{s=0}^{n-1} \gamma^s r_s, \gamma^n} \right)_\# \eta(x, a)$ the *$n$-step target distributions*. Our result shows that the back-up target of Retrace is a convex combination of $n$-step target distributions with varying values of $n$.

**Lemma 3.1. (Convex combination)** The Retrace back-up target is a convex combination of $n$-step target distributions. Formally, there exists an index set $I(x, a)$ such that $\mathcal{R}^{\pi,\mu}\eta(x, a) = \sum_{i \in I(x,a)} w_i \eta_i$ where $w_i \geq 0$, $\sum_{i \in I(x,a)} w_i = 1$ and $(\eta_i)_{i \in I(x,a)}$ are $n_i$-return target distributions.

Since $\mathcal{R}^{\pi,\mu}\eta \in \mathscr{P}_\infty(\mathbb{R})^{\mathcal{X} \times \mathcal{A}}$, we can measure the contraction of $\mathcal{R}^{\pi,\mu}$ under probability metrics.

**Proposition 3.2. (Contraction)** $\mathcal{R}^{\pi,\mu}$ is $\beta$-contractive under supremum $p$-Wasserstein distance, where $\beta = \max_{x \in \mathcal{X}, a \in \mathcal{A}} \sum_{t=1}^\infty \mathbb{E}_\mu \left[ c_1 ... c_{t-1}(1 - c_t) \right] \gamma^t \leq \gamma$.

The contraction rate of the distributional Retrace operator is determined by its effective horizon. At one extreme, when $c_t = 0$, the effective horizon is 1 and $\beta = \gamma$, in which case Retrace recovers the one-step operator. At the other extreme, when $c_t = \rho_t$, the effective horizon is infinite which gives $\beta = 0$. This latter case can be understood as correcting for all the off-policy discrepancies with IS, which is very efficient *in expectation* but incurs high variance under sample-based approximations. Proposition 3.2 also implies that the distributional Retrace operator has a unique fixed point.

**Proposition 3.3. (Unique fixed point)** $\mathcal{R}^{\pi,\mu}$ has $\eta^\pi$ as the unique fixed point in $\mathscr{P}_\infty(\mathbb{R})^{\mathcal{X} \times \mathcal{A}}$.

The above result suggests that starting with $\eta_0 \in \mathscr{P}_\infty(\mathbb{R})^{\mathcal{X} \times \mathcal{A}}$, the recursion $\eta_{k+1} = \mathcal{R}^{\pi,\mu}\eta_k$ produces iterates $(\eta_k)_{k=0}^\infty \in \mathscr{P}_\infty(\mathbb{R})^{\mathcal{X} \times \mathcal{A}}$ which converge to $\eta^\pi$ in $\overline{W}_p$ at a rate of $\mathcal{O}(\beta^k)$.

# 4 Understanding multi-step distributional reinforcement learning

Now, we pause and take a closer look at the construction of the distributional Retrace operator. We present a number of insights that distinguish distributional learning from value-based learning.

## 4.1 Path-dependent TD error

The value-based Retrace back-up target can be written as a mixture of value-based TD errors. To better parse the distributional Retrace operator and draw comparison to the value-based setting, we seek to rewrite the distributional back-up target $\mathcal{R}^{\pi,\mu}\eta(x, a)$ into a weighted sum of some notion of distributional TD errors. To this end, we start with a natural analogy to the value-based TD error.

**Definition 4.1. (Distributional TD error)** Given a transition $(X_t, A_t, R_t, X_{t+1})$, define the associated distributional TD error as $\Delta^\pi(X_t, A_t, R_t, X_{t+1}) := (b_{R_t, \gamma})_\# \eta \left( X_{t+1}, A_{t+1}^\pi \right) - \eta(X_t, A_t)$.

When the context is clear, we also adopt the concise notation $\Delta_t^\pi = \Delta^\pi(X_t, A_t, R_t, X_{t+1})$. By construction, distributional TD errors are signed measures with zero total mass [10]. The distributional TD error is a natural counterpart to the value-based TD error, because they both stem directly from the corresponding one-step Bellman operators. However, unlike in value-based RL, where TD errors alone suffice to specify the multi-step learning operator (Equation (3)), in distributional RL this is not enough. We introduce the path-dependent distributional TD error, which serves as the building block to distributional Retrace.

**Definition 4.2.** (**Path-dependent distributional TD error**) Given a trajectory $(X_s, A_s, R_s)_{s=0}^\infty$, define the path-dependent distributional TD error at time $t \geq 0$ as follows,

$$\widetilde{\Delta}_{0:t}^\pi := \left(b_{G_{0:t-1}, \gamma^t}\right)_\# \Delta_t^\pi. \tag{5}$$

Path-dependent distributional TD errors are defined as a pushforward measures from $\Delta_t^\pi$, where the pushforward operations depend on $G_{0:t-1}$. This equips $\widetilde{\Delta}_{0:t}^\pi$ with an intriguing property, *path-dependency*. Concretely, this means that the path-dependent distributional TD error depends on the sequence of rewards $(R_s)_{s=0}^{t-1}$ leading up to step $t$. With the above definitions, we can finally rewrite the back-up target of distributional Retrace as a weighted sum of path-dependent distributional TD errors $\mathcal{R}^{\pi,\mu}\eta(x,a) = \eta(x,a) + \mathbb{E}_\mu[\sum_{t=0}^\infty c_{1:t}\widetilde{\Delta}_{0:t}^\pi]$. We now illustrate the difference between value-based and distributional TD errors.

**Comparison with value-based TD equivalents.** The value-based equivalent to the path-dependent distributional TD error is the discounted value-based TD error $\widetilde{\delta}_t^\pi = \gamma^t \delta_t^\pi$ which we briefly mentioned in Section 2. To see why, note that discounted value-based TD errors allow us to rewrite the value-based Retrace back-up target as $R^{\pi,\mu}Q(x,a) = Q(x,a) + \mathbb{E}_\mu[\sum_{t=0}^\infty c_{1:t}\widetilde{\delta}_t^\pi]$. For direct comparison between the two settings, we rewrite both $\widetilde{\Delta}_{0:t}^\pi$ and $\widetilde{\delta}_t^\pi$ as the difference between two $n$-step predictions evaluated at two time steps $t$ and $t+1$,

$$\widetilde{\Delta}_{0:t}^\pi = \left(b_{G_{0:t}, \gamma^{t+1}}\right)_\# \eta\left(X_{t+1}, A_{t+1}^\pi\right) - \left(b_{G_{0:t-1}, \gamma^t}\right)_\# \eta(X_t, A_t), \qquad \text{(Distributional)}$$

$$\widetilde{\delta}_t^\pi = \left(G_{0:t} + \gamma^{t+1}Q\left(X_{t+1}, A_{t+1}^\pi\right)\right) - \left(G_{0:t-1} + \gamma^t Q(X_t, A_t)\right). \qquad \text{(Value-based)}$$

The above rewriting attributes the path-dependency to the fact that the $n$-step distributional prediction $\left(b_{G_{0:t}, \gamma^{t+1}}\right)_\# \eta\left(X_{t+1}, A_{t+1}^\pi\right)$ is non-linear in $G_{0:n-1}$. Indeed, in the value-based setting, because $G_{0:t} = G_{0:t-1} + \gamma^t R_t$ the partial sum of rewards $G_{0:t-1}$ cancels out as a common term. This leaves the discounted TD error $\widetilde{\delta}_t^\pi$ *path-independent*. In other words, the computation of $\widetilde{\delta}_t^\pi$ does not depend on past rewards $(R_s)_{s=0}^{t-1}$. In contrast, in the distributional setting, the pushforward operations are non-linear in the partial sum of rewards $G_{0:t-1}$. As a result, $G_{0:t-1}$ does not cancel out in the definition of $\widetilde{\Delta}_{0:t}^\pi$, making the path-dependent TD error $\widetilde{\Delta}_{0:t}^\pi$ depend on the past rewards $(R_s)_{s=0}^{t-1}$.

The path-dependent property is not an artifact of the distributional Retrace operator $\mathcal{R}^{\pi,\mu}$; instead, it is an indispensable element for convergent multi-step distributional learning in general. We show this by empirically verifying that multi-step learning operators based on alternative definitions of *path-independent* distributional TD errors are non-convergent even for simple problems.

**Numerically non-convergent path-independent operators.**
Consider the *path-independent* distributional TD error $\overline{\Delta}_t^\pi := (b_{0,\gamma^t})_\# \Delta_t^\pi$. We arrived at this definition by dropping the path-dependent term $G_{0:t-1}$ in the pushforward of $\widetilde{\Delta}_{0:t}^\pi$. Such a definition seems appealing because when $\eta = \eta^\pi$, the error is zero in expectation $\mathbb{E}_\mu\left[\overline{\Delta}_t^\pi | X_t, A_t\right] = 0$. This implies that we can construct a multi-step operator by a weighted sum of the alternative path-independent TD error $\overline{\mathcal{R}}_n^{\pi,\mu}\eta(x,a) := \eta(x,a) + \mathbb{E}_\mu\left[\sum_{t=0}^\infty c_{1:t}\overline{\Delta}_t^\pi\right]$. By construction, $\overline{\mathcal{R}}_n^{\pi,\mu}$ has $\eta^\pi$ as one fixed point.

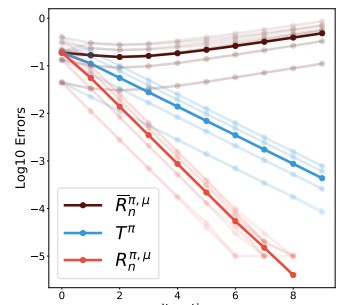

We provide a very simple counterexample on which $\overline{\mathcal{R}}^{\pi,\mu}$ is not contractive: consider an MDP with one state and one action. The state transitions back to itself with a deterministic reward $R_t = 1$. When the discount factor is $\gamma = 0.5$, $\eta^\pi$ is a Dirac distribution centered at 2. We consider the simple case $c_1 = \rho_1$ and $c_t = 0, \forall t \geq 2$. We use the $L_p$ distance to measure the

Figure 2: Non-convergent example: comparing $L_p(\mathcal{R}^k\eta_0, \eta^\pi)$ across iterations. We plot 10 randomly initialized runs. Note $(\overline{\mathcal{R}}^{\pi,\mu})^k\eta_0$ does not converge to $\eta^\pi$ while others do.

convergence of the distribution iterates [10]. Figure 2 shows that $(\overline{\mathcal{R}}_n^{\pi,\mu})^k\eta_0$ does not converge to $\eta^\pi$, while the one-step Bellman operator $\mathcal{T}^\pi$ and distributional Retrace $\mathcal{R}^{\pi,\mu}$ are convergent.

In Appendix C, we discuss yet another alternative to $\widetilde{\Delta}_{0:t}^\pi$ designed to be path-independent $\gamma^t \Delta_t^\pi$. Though the resulting multi-step operator still has $\eta^\pi$ as one fixed point, we show numerically that it

is not contractive on the same simple example. These results demonstrate that naively removing the path-dependency might lead to non-convergent multi-step operators.

## 4.2 Backward-view of distributional multi-step learning

To highlight the difference between distributional and value-based multi-step learning, we discuss the impact that path-dependent distributional TD errors have on the backward-view distributional algorithm. Thus far, distributional back-up targets are expressed in the *forward-view*, i.e., the back-up target at time $t$ is calculated as a function of future transition tuples $(X_s, A_s, R_s)_{s \leq t}$. The forward-view algorithms, unless truncated, wait until the episode finishes to carry out the update, which might be undesirable when the problem is non-episodic or has a very long horizon.

In the *backward-view*, when encountering a distributional TD error $\Delta_t^\pi$, the algorithm carries out updates for all predictions at time $t' \leq t$ [2]. To this end, the algorithm needs to maintain additional *partial return traces*, i.e., the partial sum of rewards $G_{t':t}$, in order to calculate the path-dependent TD error $\widetilde{\Delta}_t^\pi$. Unlike the value-based state-dependent eligibility traces [2, 18], partial return traces are time-dependent. This implies that in an episode of $T$ steps, value-based backward-view algorithms require memory of size $\min(|\mathcal{X}||\mathcal{A}|, \mathcal{O}(T))$ while the distributional algorithms requires $\mathcal{O}(T)$.

In addition to the added memory complexity, the incremental updates of distributional algorithms are also much more complicated due to the path-dependent TD errors. We remark that the path-independent nature of value-based TD errors greatly simplify the value-based backward-view algorithm. For a more detailed discussion, see Appendix D.

## 4.3 Importance sampling for multi-step distributional RL

In our initial derivation, we arrived at $\mathcal{R}^{\pi,\mu}$ through the application of importance sampling (IS) in a different way from the value-based setting. We now highlight the subtle differences and caveats.

For a fixed $n \geq 1$, consider the trace coefficient $c_t = \rho_t \mathbb{I}[t < n]$. The back-up target of the resulting Retrace operator reduces to $\mathbb{E}_\mu \left[ \rho_{1:n-1} \cdot \left( \mathrm{b}_{G_{0:n-1}, \gamma^n} \right)_\# \eta \left( X_n, A_n^\pi \right) \right]$. This can be seen as applying IS to the $n$-step prediction $\left( \mathrm{b}_{G_{0:n-1}, \gamma^n} \right)_\# \eta \left( X_n, A_n^\pi \right)$. As a caveat, note that an appealing alternative approach is to apply IS to $G_{0:n-1}$, producing the estimate $\left( \mathrm{b}_{\rho_{1:n-1} G_{0:n-1}, \gamma^n} \right)_\# \eta \left( X_n, A_n^\pi \right)$. This latter estimate does not properly correct for the off-policy discrepancy between $\pi$ and $\mu$. To see why, note that applying the IS ratio to $G_{0:n-1}$, instead of to the probability of its occurrence, is an artifact of value-based RL because the expected return is linear in $G_{0:t}$ [11]. In general for distributional RL, one should importance weigh the measures instead of sum of rewards.

# 5 Approximate multi-step distributional reinforcement learning algorithm

We now discuss how the distributional Retrace operator combines with parametric distributions, using the construction of the novel Quantile Regression-Retrace algorithm as a practical example. We focus on the quantile representation because it entails the best empirical performance of large-scale distributional RL [16, 19]. Speficially, we present an application of quantile regression with signed measures, which is interesting in its own right. Below, we start with a brief background on quantile representations [16], followed by details on the proposed algorithm.

Consider parametric distributions of the form: $\frac{1}{m} \sum_{i=1}^{m} \delta_{z_i}$ for a fixed $m \geq 1$, where $(z_i)_{i=1}^m \in \mathbb{R}$ are a set of parameters indicating the support of the distribution. Let $\mathscr{P}_\mathcal{Q}(\mathbb{R})$ denote the family of distribution $\mathscr{P}_\mathcal{Q}(\mathbb{R}) := \{ \frac{1}{m} \sum_{i=1}^{m} \delta_{z_i} | z_i \in \mathbb{R} \}$. We define the projection $\Pi_\mathcal{Q} : \mathscr{P}_\infty(\mathbb{R}) \to \mathscr{P}_\mathcal{Q}(\mathbb{R})$ as $\Pi_\mathcal{Q} \eta = \arg \min_{\nu \in \mathscr{P}_\mathcal{Q}(\mathbb{R})} W_1(\eta, \nu)$, which projects any distribution onto the space of representable distributions in the parametric class under the $W_1$ distance. With an abuse of notation, we also let $\Pi_\mathcal{Q}$ denote the component-wise projection when applied to vectors. See [16, 10] for more details.

**Gradient-based learning via quantile regression.** We can use quantile regression [20–22] to calculate the projection $\Pi_\mathcal{Q} \eta$. Let $F_\eta(z), z \in \mathbb{R}$ denote the CDF of a given distribution $\eta$. Let $F_\eta^{-1}$ be the generalized CDF inverse, we define the $\tau$-th quantile as $F_\eta^{-1}(\tau)$ for $\tau \in [0, 1]$. The projection $\Pi_\mathcal{Q}$ is equivalent to computing $z_i = F_\eta^{-1}(\tau_i)$ for $\tau \in (\frac{2i-1}{2m})_{i=1}^m$ [16]. To learn the $\tau$-th quantile for any $\tau \in [0, 1]$, it suffices to solve the quantile regression problem whose optimal solution is $F_\eta^{-1}(\tau)$: $\min_\theta L_\theta^\tau(\eta) := \mathbb{E}_{Z \sim \eta} [f_\tau(Z - \theta)]$ where $f_\tau(u) = u(\tau - \mathbb{I}[u < 0])$. In practice, we carry out the gradient update $\theta \leftarrow \theta - \alpha \nabla_\theta L_\theta^\tau(\eta)$ to find the optimal solution and learn the quantile $\theta \approx F_\eta^{-1}(\tau)$.

## 5.1 Distributional Retrace with quantile representations

Given an input distribution vector $\eta$, we use the distributional Retrace operator to construct the back-up target $\mathcal{R}^{\pi,\mu}\eta$. Then, we use the quantile projection to map the back-up target onto the space of representations $\Pi_{\mathcal{Q}}\mathcal{R}^{\pi,\mu}\eta$. Overall, we are interested in the recursive update: start with any $\eta_0 \in \mathscr{P}_{\mathcal{Q}}(\mathbb{R})^{\mathcal{X}\times\mathcal{A}}$, consider the sequence of distributions generated via $\eta_{k+1} = \Pi_{\mathcal{Q}}\mathcal{R}^{\pi,\mu}\eta_k$. A direct application of Proposition 3.2 allows us to characterize the convergence of the sequence, following the approach of [10].

**Theorem 5.1.** (**Convergence of quantile distributions**) The projected distributional Retrace operator $\Pi_{\mathcal{Q}}\mathcal{R}^{\pi,\mu}$ is $\beta$-contractive under $\overline{W}_\infty$ distance in $\mathscr{P}_{\mathcal{Q}}(\mathbb{R})$. As a result, the above $\eta_k$ converges to a limiting distribution $\eta_{\mathcal{R}}^\pi$ in $\overline{W}_\infty$, such that $\overline{W}_\infty(\eta_k, \eta_{\mathcal{R}}^\pi) \le (\beta)^k \overline{W}_\infty(\eta_0, \eta_{\mathcal{R}}^\pi)$. Further, the quality of the fixed point is characterized as $\overline{W}_\infty(\eta_{\mathcal{R}}^\pi, \eta^\pi) \le (1-\beta)^{-1}\overline{W}_\infty(\Pi_{\mathcal{Q}}\eta^\pi, \eta^\pi)$.

Thanks to the faster contraction rate $\beta \le \gamma$, the advantage of the projected operator $\Pi_{\mathcal{Q}}\mathcal{R}^{\pi,\mu}$ is two-fold: (1) the operator often contracts faster to the limiting distribution $\eta_{\mathcal{R}}^\pi$ than the one-step operator $\mathcal{T}^\pi$ contracts to its own limiting distribution $\eta_{\mathcal{T}^\pi}$ [16]; (2) the limiting distribution $\eta_{\mathcal{R}}^\pi$ also enjoys a better approximation bound to the target distribution. We verify such results in Section 7.

## 5.2 Quantile Regression-Retrace: distributional Retrace with quantile regression

Below, we use $z_i(x,a)$ to represent the $i$-th quantile of the distribution at $(x,a)$. Overall, we have a tabular quantile representation $\eta_z(x,a) = \frac{1}{m}\sum_{i=1}^m \delta_{z_i(x,a)}, \forall(x,a) \in \mathcal{X}\times\mathcal{A}$, where we use the notation $\eta_z$ to stress the distribution's dependency on parameter $z_i(x,a)$. For any given bootstrapping distribution vector $\eta \in \mathscr{P}_\infty(\mathbb{R})^{\mathcal{X}\times\mathcal{A}}$, in order to approximate the projected back-up target $\Pi_{\mathcal{Q}}\mathcal{R}^{\pi,\mu}\eta$ with the parameterized quantile distribution $\eta_z$, we solve the set of quantile regression problems for all $1 \le i \le m, (x,a) \in \mathcal{X}\times\mathcal{A}$,

$$\min_{z_i(x,a)} L_{z_i(x,a)}^{\tau_i}\left(\mathcal{R}^{\pi,\mu}\eta(x,a)\right), \text{ where } \tau_i = (2i-1)/2m.$$

For any fixed $(x,a,i)$, to solve the quantile regression problem, we apply gradient descent on $z_i(x,a)$. In practice, with one sampled trajectory $(X_s, A_s, R_s)_{s=0}^\infty \sim \mu$, the aim is to construct an unbiased stochastic gradient estimate of the QR loss $L_{z_i(x,a)}^{\tau_i}(\mathcal{R}^{\pi,\mu}\eta(x,a))$. Below, let $\mathsf{b}_t = \mathsf{b}_{G_{0:t-1},\gamma^t}$ for simplicity. We start with a stochastic estimate $\widehat{L}_{z_i(x,a)}^{\tau_i}(\mathcal{R}^{\pi,\mu}\eta(x,a))$ for the QR loss,

$$L_{z_i(x,a)}^{\tau_i}\left(\eta(x,a)\right) + \sum_{t=0}^\infty c_{1:t}\left(L_{z_i(x,a)}^{\tau_i}\left((\mathsf{b}_{t+1})_\# \eta\left(X_{t+1}, A_{t+1}^\pi\right)\right) - L_{z_i(x,a)}^{\tau_i}\left((\mathsf{b}_t)_\# \eta(X_t, A_t)\right)\right).$$

Since $\widehat{L}_{z_i(x,a)}^{\tau_i}(\mathcal{R}^{\pi,\mu}\eta(x,a))$ is differentiable with $z_i(x,a)$, we use $\nabla_{z_i(x,a)}\widehat{L}_{z_i(x,a)}^{\tau_i}(\mathcal{R}^{\pi,\mu}\eta(x,a))$ as the stochastic gradient estimate. This gradient estimate is unbiased under mild conditions.

**Lemma 5.2.** (**Unbiased stochastic QR loss gradient estimate**) Assume that the trajectory terminates within $H < \infty$ steps almost surely, then we have $\mathbb{E}_\mu[\widehat{L}_{z_i(x,a)}^{\tau_i}(\mathcal{R}^{\pi,\mu}\eta(x,a))] = L_{z_i(x,a)}^{\tau_i}(\mathcal{R}^{\pi,\mu}\eta(x,a))$ and $\mathbb{E}_\mu[\nabla_{z_i(x,a)}\widehat{L}_{z_i(x,a)}^{\tau_i}(\mathcal{R}^{\pi,\mu}\eta(x,a))] = \nabla_{z_i(x,a)}L_{z_i(x,a)}^{\tau_i}(\mathcal{R}^{\pi,\mu}\eta(x,a))$.

The above stochastic estimate bypasses the challenge that the QR loss is only defined against distributions, whereas sampled back-up targets $\widehat{\mathcal{R}}^{\pi,\mu}\eta(x,a) = \eta(x,a) + \sum_{t=0}^\infty c_{1:t}\widetilde{\Delta}_{0:t}^\pi$ are signed measures in general. In Quantile Regression-Retrace, we use $\eta_z$ itself as the bootstrapping distribution, such that the algorithm approximates the fixed point iteration $\eta_z \leftarrow \Pi_{\mathcal{Q}}\mathcal{R}^{\pi,\mu}\eta_z$. Concretely, we carry out the following sample-based update

$$z_i(x,a) \leftarrow z_i(x,a) - \alpha\nabla_{z_i(x,a)}\widehat{L}_{z_i(x,a)}^{\tau_i}(\mathcal{R}^{\pi,\mu}\eta_z(x,a)), \text{ for } \forall 1 \le i \le m, (x,a) \in \mathcal{X}\times\mathcal{A}.$$

## 5.3 Deep reinforcement learning: QR-DQN-Retrace

We introduce a deep RL implementation of the Quantile Regression-Retrace: QR-DQN-Retrace, where the parametric representation is combined with function approximations [23, 16, 19]. The base agent QR-DQN [23] parameterizes the quantile locations $z_i(x,a;w)$ with the output of a neural network with weights $w$. Let $\eta(x,a;w) = \frac{1}{m}\sum_{i=1}^m \delta_{z_i(x,a;w)}$ denote the parameterized distribution. QR-DQN-Retrace updates its parameters by stochastic gradient descent on the estimated QR loss, averaged across all $m$ quantile levels $w \leftarrow w - \alpha\frac{1}{m}\sum_{i=1}^m \nabla_w \widehat{L}_{z_i(x,a;w)}^{\tau_i}(\mathcal{R}^{\pi,\mu}\eta(x,a;w))$. In practice, the update is further averaged over state-action pairs sampled from a replay buffer.

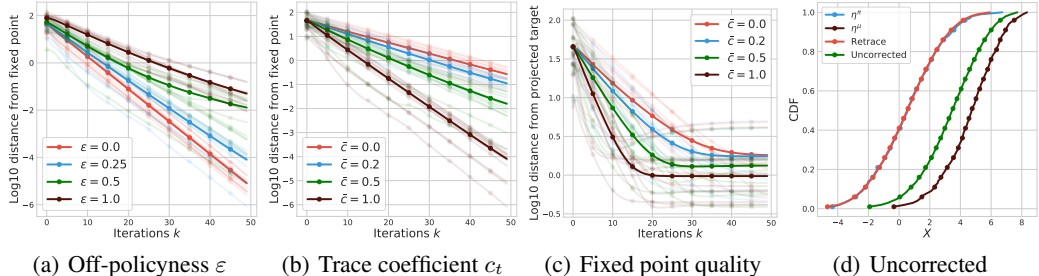

| (a) Off-policyness $\varepsilon$ | (b) Trace coefficient $c_t$ | (c) Fixed point quality | (d) Uncorrected |

Figure 3: Tabular experiments to illustrate properties of the distributional Retrace operator: we show average results across 10 randomly sampled MDPs. (a) Contraction rate vs. off-policyness; (b) Contraction rate vs. trace coefficient $c_t = \min(\rho_t, \bar{c})$; (c) Fixed point quality vs. trace coefficient $c_t$; (d) The uncorrected operator introduces bias to the fixed point while Retrace is unbiased.

## 6 Discussions

**Categorical representations.** The categorical representation is another commonly used class of parameterized distributions in prior literature [23, 17, 24, 10]. We obtain contractive guarantees for the categorical representation similar to Theorem 5.1. As with QR, this leads both to improved fixed-point approximations and faster convergence. Further, this leads to a deep RL algorithm C51-Retrace. The actor-critic Reactor agent [25] uses C51-Retrace as a critic training algorithm, although without explicit consideration or analysis of the associated distributional operator. See Appendix E for details. We empirically evaluate the stand-alone improvements of C51-Retrace over C51 in Section 7.

**Uncorrected methods.** The uncorrected methods do not correct for the off-policyness and hence obtain a biased fixed point [26–28]. The Rainbow agent [26] combined $n$-step uncorrected learning with C51, effectively implementing a distributional operator whose fixed point differs from $\eta^\pi$.

**On-policy distributional TD($\lambda$).** Nam et al. [29] propose SR($\lambda$), a distributional version of on-policy TD($\lambda$) [30]. In operator form, this can be viewed as a special case of Equation (4) with $\mu = \pi$, $c_t = \lambda$; [29] also introduce a sample-replacement technique for more efficient implementation.

## 7 Experiments

We carry out a number of experiments to validate the theoretical insights and empirical improvements.

### 7.1 Illustration of distributional Retrace properties on tabular MDPs

We verify a few important properties of the distributional Retrace operator on a tabular MDP. The results corroborate the theoretical results from previous sections. Throughout, we use quantile representations with $m = 100$ atoms; we obtain similar results for categorical representations. See Appendix F for details on the experiment setup. Let $\eta_0$ be the initial distribution, we carry out dynamic programming with $\mathcal{R}^{\pi,\mu}$ and denote $\eta_k = (\mathcal{R}^{\pi,\mu})^k \eta_0$ as the $k^{\text{th}}$ distribution iterate.

**Impact of off-policyness.** We control the level of off-policyness by setting the behavior policy $\mu$ to be a uniform policy and the target policy to $\pi = (1 - \varepsilon)\mu + \varepsilon\pi_d$ where $\pi_d$ is a fixed deterministic policy. Moving from $\varepsilon = 0$ to $\varepsilon = 1$, we transition from on-policy to very off-policy. We use $L_p(\eta_k, \eta_{\mathcal{R}}^\pi)$ to measure the contraction rate to the fixed point. Figure 3 shows that as the behavior becomes more off-policy, the contraction slows down, degrading the efficiency of multi-step learning.

**Impact of trace coefficient $c_t$.** Throughout, we set $c_t = \min(\rho_t, \bar{c})$ with $\bar{c}$ to control the effective trace length. With a fixed level of off-policyness $\varepsilon = 0.5$, Figure 3(b) shows that increasing $\bar{c}$ speeds up the contraction to the fixed point as predicted by Proposition 3.2.

**Quality of fixed point.** We next examine how the quality of the fixed point is impacted by $\bar{c}$, by measuring $L_p(\eta_k, \Pi_{\mathcal{Q}}\eta^\pi)$ as a proxy to $L_p(\eta_k, \eta^\pi)$. As $k$ increases the error flattens, at which point we take the converged value to be $L_p(\eta_{\mathcal{R}}^\pi, \Pi_{\mathcal{Q}}\eta^\pi)$ which measures the fixed point quality. Figure 3(c) shows when $\bar{c}$ increases, the fixed point quality improves, in line with the Theorem 5.1. This phenomenon does not arise in *tabular* non-distributional reinforcement learning, although related phenomena do occur when using function approximation techniques.

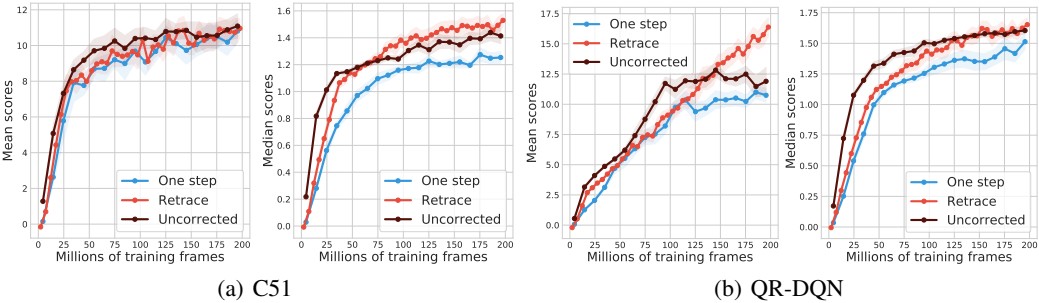

|  (a) C51 | (b) QR-DQN |

Figure 4: Deep RL experiments on Atari-57 games for (a) C51 and (b) QR-DQN. We compare the one-step baseline agent against the multi-step variants (Retrace and uncorrected $n$-step). For all multi-step variants, we use $n = 3$. For each agent, we calculate the mean and median performance across all games, and we plot the mean $\pm$ standard error across 3 seeds. In almost all settings, multi-step variants provide clear advantage over the one-step baseline algorithm.

**Bias of uncorrected methods.** Finally, we illustrate a critical difference between Retrace and uncorrected $n$-step methods [26]: the bias to the fixed point. Figure 3(d) shows that uncorrected $n$-step arrives at a fixed point in between $\eta^\pi$ and $\eta^\mu$, showing an obvious bias from $\eta^\pi$.

### 7.2 Deep reinforcement learning

We consider the control setting where the target policy $\pi$ is the greedy policy with respect to the Q-function induced by the parameterized distribution. Because the training data is sampled from a replay, the behavior policy $\mu$ is $\varepsilon$-greedy with respect to Q-functions induced by previous copies of the parameterized distribution. We evaluate the performance of deep RL agents on 57 Atari games [31]. To ensure fair comparison across games, we compute the human normalized scores for each agent, and compare their evaluated mean and median scores across all 57 games during training.

**Deep RL agents.** The multi-step agents adopt exactly the same hyperparameters as the baseline agents. The only difference is the back-up target. For completeness of results, we show the combination of Retrace with both C51 and QR-DQN. For QR-DQN, we use the Huber loss for quantile regression, which is a thresholded variant of the QR loss [16]. Throughout, we use $c_t = \lambda \min(\rho_t, \bar{c})$ with $\bar{c} = 1$ as in [13]. See Appendix F for details. In practice, sampled trajectories are truncated at length $n$. We also adapt Retrace to the $n$-step case, see Appendix A.

**Results.** Figure 4 compares one-step baseline, Retrace and uncorrected $n$-step [26]. For C51, both multi-step methods clearly improve the median performance over the one-step baseline. Retrace slightly outperforms uncorrected $n$-step towards the end of learning. For QR-DQN, all multi-step algorithms achieve clear performance gains. Retrace significantly outperforms the uncorrected $n$-step with the mean performance, while obtaining similar results on the median performance. Overall, distributional Retrace achieves a clear improvement over the one-step baselines. The uncorrected $n$-step method typically takes off faster than Retrace but may to slightly worse performance.

Finally, note that in the value-based setting, uncorrected methods are generally more high-performing than Retrace, potentially due to a favorable trade-off between contraction rate and fixed-point bias [32]. Our results add to the benefits of off-policy corrections in the control setting.

## 8 Conclusion

We have identified a number of fundamental conceptual differences between value-based and distributional RL in multi-step settings. Central to such differences is the novel notion of path-dependent distributional TD error, which naturally arises from the multi-step distributional RL problem. Building on this understanding, we have developed the first principled multi-step off-policy distributional operator Retrace. We have also developed an approximate distributional RL algorithm, Quantile Regression-Retrace, which makes distributional Retrace highly competitive in both tabular and high-dimensional setups. This paper also opens up a several avenues for future research, such as the interaction between multi-step distributional RL and signed measures, and the convergence theory of stochastic approximations for multi-step distributional RL.

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
