# The Nature of Temporal Difference Errors in Multi-step Distributional Reinforcement Learning: Appendices

## A  Extension of distributional Retrace to $n$-step truncated trajectories

The $n$-step truncated version of distributional Retrace is defined as

$$\mathcal{R}_n^{\pi,\mu}\eta(x,a) = \eta(x,a) + \mathbb{E}_\mu\left[\sum_{t=0}^n c_{1:t}\widetilde{\Delta}_{0:t}^\pi\right],$$

which sums the path-dependent distributional TD errors up to time $n$. Compared to the original definition of distributional Retrace, this $n$-step operator is more practical to implement. This operator enjoys all the theoretical properties of the original distributional Retrace, with a slight difference on the contraction rate. Intuitively, the operator bootstraps with at most $n$ steps, which limits the effective horizon of the operator to be $\leq n$. It is straightforward to show that the operator is $\beta_n$-contractive under $\overline{W}_p$ with $\beta_n \in (\beta, \gamma]$. As $n \to \infty$, $\beta_n \to \beta$.

## B  Distance metrics

We provide a brief review on the distance metrics used in this work. We refer readers to [10] for a complete background.

### B.1  Wasserstein distance

Let $\eta_1, \eta_2 \in \mathscr{P}_\infty(\mathbb{R})$ be two distribution measures. Let $F_\eta$ be the CDF of $\eta$. The $p$-Wasserstein distance can be computed as

$$W_p(\eta_1, \eta_2) := \left(\int_{[0,1]} |F_{\eta_1}^{-1}(z) - F_{\eta_2}^{-1}(z)|^p dz\right)^{1/p}.$$

Note that the above definition is equivalent to the more traditional definition based on optimal transport; indeed, $F_{\eta_i}^{-1}(z), z \sim \text{Uniform}(0,1), i \in \{1,2\}$ can be understood as the optimal coupling between the two distributions. The above definition is a proper distance metric if $p \geq 1$.

For any distribution vector $\eta_1, \eta_2 \in \mathscr{P}_\infty(\mathbb{R})^{\mathcal{X}\times\mathcal{A}}$, we can define the supremum $p$-Wasserstein distance as

$$\overline{W}_p(\eta_1, \eta_2) := \max_{x,a} W_p(\eta_1(x,a), \eta_2(x,a)).$$

### B.2  $L_p$ distance

Let $\eta_1, \eta_2 \in \mathscr{P}_\infty(\mathbb{R})$ be two distribution measures. Let $F_\eta$ be the CDF of $\eta$. The $L_p$ distance is defined as

$$L_p(\eta_1, \eta_2) := \left(\int_{\mathbb{R}} |F_{\eta_1}(z) - F_{\eta_2}(z)|^p dz\right)^{1/p}.$$

The above definition is a proper distance metric when $p \geq 1$.

For any distribution vector $\eta_1, \eta_2 \in \mathscr{P}_\infty(\mathbb{R})^{\mathcal{X}\times\mathcal{A}}$ or signed measure vector $\eta_1, \eta_2 \in \mathcal{M}(\mathbb{R})^{\mathcal{X}\times\mathcal{A}}$, we can define the supremum Cramér-$p$ distance as

$$\overline{L}_p(\eta_1, \eta_2) := \max_{x,a} L_p(\eta_1(x,a), \eta_2(x,a)).$$

## C  Numerically non-convergent behavior of alternative multi-step operators

We consider another alternative definition of path-independent alternative to the path-dependent TD error $\gamma^t \Delta_t^\pi$. The primary motivation for such a path-dependent TD error is that the discounted value-based TD error takes the form $\widetilde{\delta}_t^\pi = \gamma^t \delta_t^\pi$. The resulting multi-step operator is

$$\widetilde{\mathcal{R}}^{\pi,\mu}\eta(x,a) = \eta(x,a) + \mathbb{E}_\mu\left[\sum_{t=0}^\infty c_{1:t}\gamma^t \Delta_t^\pi\right].$$

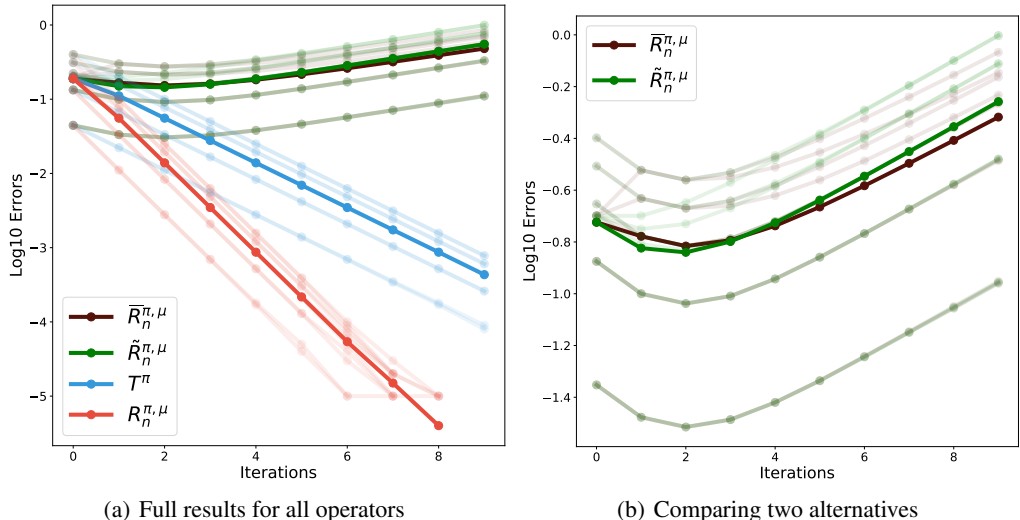

(a) Full results for all operators

(b) Comparing two alternatives

Figure 5: Illustration of non-convergent behavior of alternative multi-step operators: for both plots, we show the mean and per-run results across 10 different initial Dirac distributions $\eta_0$. (a) the full comparison between all operators. Two alternative operators do not converge while one-step Bellman operator and distributional Retrace both converge; (b) we zoom in on the difference between the two alternative operators.

With the same toy example as in the paper: an one-state one-action MDP with a deterministic reward $R_t = 1$ and discount factor $\gamma = 0.5$. The target distribution $\eta^\pi$ is a Dirac distribution centering at 2. Let $\eta_k = (\mathcal{R})^k \eta_0$ be the $k$-th distribution iterate by applying the operator $\mathcal{R} \in \{\mathcal{R}^{\pi,\mu}, \widetilde{\mathcal{R}}^{\pi,\mu}, \widetilde{\mathcal{R}}^{\pi,\mu}, \mathcal{T}^\pi\}$, we show the $L_p$ distance between the iterates and $\eta^\pi$ in Figure 5. It is clear that alternative multi-step operators do not converge to the correct fixed point.

## D  Backward-view algorithm for multi-step distributional RL

We now describe a backward-view algorithm for multi-step distributional RL with quantile representations. For simplicity, we consider the on-policy case $\pi = \mu$ and $c_t = \lambda$. To implement $\mathcal{R}^{\pi,\mu}$ in the backward-view, at each time step $t$ and a past time step $t' \leq t$, the algorithm needs to maintain two novel traces distinct from the classic eligibility traces [2]: (1) partial return traces $G_{t':t}$, which correspond to the partial sum of rewards between two time steps $t' \leq t$; (2) modified eligibility traces, defined as $e_{t',t} := \lambda^{t-t'}$, which measures the trace decay between two time steps $t' \leq t$. At a new time step $t + 1$, the new traces are computed recursively: $G_{t':t+1} = R_{t+1} + \gamma G_{t':t}, e_{t',t+1} = \lambda e_{t',t}$.

We assume the algorithm maintains a table of quantile distributions with $m$ atoms: $\eta(x,a) = \frac{1}{m} \sum_{i=1}^{m} \delta_{z_i(x,a)}, \forall (x,a) \in \mathcal{X} \times \mathcal{A}$. For any fixed $(x,a)$, define $T_t(x,a) := \{s | X_s = x, A_s = a, 0 \leq s \leq t\}$ be the set of time steps before time $t$ at which $(x,a)$ is visited. Now, upon arriving at $X_{t+1}$, we observe the TD error $\Delta_t^\pi$. Recall that $L_\theta^\tau(\eta)$ denote the QR loss of parameter $\theta$ at quantile level $\tau$ and against the distrbution $\eta$. To more conveniently describe the update, we define the QR loss against the path-dependent TD error

$$\left(b_{G_{s:t-1}, \gamma^{t-s}}\right)_{\#} \widetilde{\Delta}_{0:t}^\pi = \left(b_{G_{s:t}, \gamma^{t+1-s}}\right)_{\#} \eta(X_{t+1}, A_{t+1}^\pi) - \left(b_{G_{s:t-1}, \gamma^{t-s}}\right)_{\#} \eta(X_t, A_t)$$

as the difference of the QR losses against the individual distributions,

$$L_\theta^\tau \left(\left(b_{G_{s:t-1}, \gamma^{t-s}}\right)_{\#} \widetilde{\Delta}_{0:t}^\pi\right) := L_\theta^\tau \left(\left(b_{G_{s:t}, \gamma^{t+1-s}}\right)_{\#} \eta(X_{t+1}, A_{t+1}^\pi)\right) - L_\theta^\tau \left(\left(b_{G_{s:t-1}, \gamma^{t-s}}\right)_{\#} \eta(X_t, A_t)\right).$$

Note that the QR loss can be computed using the transition data we have seen so far. We now perform the a gradient update for all entries in the table $(x,a) \in \mathcal{X} \times \mathcal{A}$ and $1 \leq i \leq m$ (in practice, we update entries that correspond to visited state-action pairs):

$$z_i(x,a) \leftarrow z_i(x,a) - \alpha \sum_{s \in T_t(x,a)} e_{s,t} \nabla_{z_i(x,a)} L_\theta^{\tau_i} \left(\left(b_{G_{s:t-1}, \gamma^{t-s}}\right)_{\#} \widetilde{\Delta}_{0:t}^\pi\right),$$

where $\tau_i = \frac{2i-1}{2m}$. For any fixed $(x,a)$, the above algorithm effectively aggregates updates from time steps $s \in T_t(x,a)$ at which $(x,a)$ is visited.

## D.1 Simplifications for value-based RL

We now discuss how the path-independent value-based TD errors greatly simplify the value-based backward-view algorithm. Following the above notations, assume the algorithm maintains a table of Q-function $Q(x,a)$, we can construct incremental backward-view update for all $(x,a) \in \mathcal{X} \times \mathcal{A}$ as follows, by replacing the path-dependent distributional TD error $\widetilde{\Delta}_{0:t}^\pi$ by the discounted TD error $\widetilde{\delta}_t^\pi$

$$Q(x,a) \leftarrow Q(x,a) - \alpha \sum_{s \in T_t(x,a)} e_{s,t} \widetilde{\delta}_t^\pi.$$

Since $\delta_t^\pi$ does not depend on the past rewards and is state-action dependent, we can simplify the summation over $s \in T_t(x,a)$ by defining the state-depedent eligibility traces [2] as a replacement to $e_{s,t}$,

$$\widetilde{e}(x,a) \leftarrow \gamma\lambda\widetilde{e}(x,a) + \mathbb{I}[X_t = x, A_t = a].$$

As a result, the above update reduces to

$$Q(x,a) \leftarrow Q(x,a) - \alpha\widetilde{e}(x,a)\delta_t^\pi,$$

which recovers the classic backward-view update.

## D.2 Non-equivalence of forward-view and backward-view algorithms

In value-based RL, forward-view and backward-view algorithms are equivalent given that the trajectory does not visit the same state twice [2]. However, such an equivalence does not generally hold in distributional RL. Indeed, consider the following counterexample in the case of the quantile representation.

Consider a three-step MDP with deterministic transition $x_1 \rightarrow x_2 \rightarrow x_3$. There is no action and no reward on the transition. The state $x_3$ is terminal with a deterministic terminal value $r_3 = 1$. We consider $m = 1$ atom and let the quantile parameters be $\theta_1 = 0$ and $\theta_2 = 1$ at states $x_1, x_2$ respectively. In this case, the quantile representation learns the median of the target distribution with $\tau = 0.5$.

Now, we consider the update at $\theta_1$ with both forward-view and backward-view implementation of the two-step Bellman operator $\mathcal{T}_2^\pi \eta(x) = \mathbb{E}\left[(b_{0,\gamma^2})_{\#}\eta(X_2, \pi(X_2))|X_0 = x\right]$, which can be obtained from distributional Retrace by setting $c_t = \rho_t$. The target distribution at $x_1$ is a Dirac distribution centering at $\gamma^2$.

**Forward-view update.** Below, we use $\delta_x$ to denote a Dirac distribution at $x$. In the forward-view, the back-up distribution is

$$\mathbb{E}\left[(b_{0,\gamma^2})_{\#}\eta(X_2, \pi(X_2))\right] = \delta_{\gamma^2}.$$

The gradient update to $\theta_1$ is thus

$$\theta_1^{(\text{fwd})} = \theta_1 - \alpha\nabla_{\theta_1}L_{\theta_1}^{0.5}\left(\delta_{\gamma^2}\right) = \theta_1 + \alpha\left(0.5 - \mathbb{I}\left[\gamma^2 < \theta_1\right]\right).$$

**Backward-view update.** To implement the backward-view update, we make clear of the two path-dependent distributional TD errors at two consecutive time steps

$$\widetilde{\Delta}_0^\pi = \delta_\gamma - \delta_0, \quad \widetilde{\Delta}_1^\pi = (b_{0,\gamma})_{\#}\left(\delta_{\gamma\theta_2} - \delta_{\theta_1}\right) = \delta_{\gamma^2} - \delta_\gamma$$

The update consists of two steps:

$$\theta_1' = \theta_1 - \alpha\nabla_{\theta_1}L_{\theta_1}^{0.5}\left(\delta_\gamma\right) = \theta_1 + \alpha\left(0.5 - \mathbb{I}\left[\gamma < \theta_1\right]\right),$$

$$\theta_1^{(\text{bwd})} = \theta_1' - \alpha\left(\nabla_{\theta_1'}L_{\theta_1'}^{0.5}\left(\delta_\gamma^2\right) - \nabla_{\theta_1'}L_{\theta_1'}^{0.5}\left(\delta_\gamma\right)\right)$$

$$= \theta_1' + \alpha\left(0.5 - \mathbb{I}[\gamma^2 < \theta_1']\right) - \alpha\left(0.5 - \mathbb{I}[\gamma < \theta_1']\right).$$

Overall, we have

$$\theta_1^{(\text{bwd})} = \theta_1 + \alpha\left(0.5 - \mathbb{I}\left[\gamma < \theta_1\right]\right) + \alpha\left(0.5 - \mathbb{I}[\gamma^2 < \theta_1']\right) - \alpha\left(0.5 - \mathbb{I}[\gamma < \theta_1']\right)$$

$$= 0.5\alpha - \alpha\mathbb{I}[\gamma^2 < 0.5\alpha] + \mathbb{I}[\gamma < 0.5\alpha].$$

Now, let $\alpha \in (2\gamma^2, 2\gamma)$ such that $0.5\alpha \in (\gamma^2, \gamma)$, we have $\theta_1^{\text{bwd}} = 0.5\alpha - \alpha = -0.5\alpha \neq \theta_1^{(\text{fwd})}$.

### D.3 Discussion on memory complexity

The return traces $G_{t',t}$ and modified eligibility traces $e_{t',t}$ are time-dependent, which is a direct implication from the fact that distributional TD errors are path-dependent. Indeed, to calculate the distributional TD error $\widetilde{\Delta}_{t':t}^\pi$, it is necessary to keep track $G_{t',t}$ in the backward-view algorithm. This differs from the classic eligibility traces, which are state-action-dependent [2, 18]. We remark that the state-action-dependency of eligibility traces result from the fact that value-based TD errors $\Delta_t^\pi$ are path-independent. The time-dependency greatly influences the memory complexity of the algorithm: when an episode is of length $T$, value-based backward-view algorithm requires memory of size $\min(|\mathcal{X}||\mathcal{A}|, T)$ to store all eligibility traces. On the other hand, the distributional backward-view algorithm requires $\mathcal{O}(T)$.

## E  Distributional Retrace with categorical representations

We start by showing that the distributional Retrace operator is $\beta_{L_p}$-contractive under the $\overline{L}_p$ distance for $p \geq 1$. As a comparison, the one-step distributional Bellman operator $\mathcal{T}^\pi$ is $\gamma^{1/p}$-contractive under $\overline{L}_p$ [17].

**Lemma E.1.** (**Contraction in $\overline{L}_p$**) $\mathcal{R}^{\pi,\mu}$ is $\beta_{L_p}$-contractive under supremum $L_p$ distance for $p \geq 1$, where $\beta_{L_p} \in [0, \gamma]$. Specifically, we have $\beta_{L_p} = \max_{x \in \mathcal{X}, a \in \mathcal{A}} \left( \sum_{t=1}^\infty \mathbb{E}_\mu \left[ c_1 ... c_{t-1}(1 - c_t) \right] \gamma^t \right)^{1/p}$.

*Proof.* The proof is similar to the proof of Proposition 3.2: the result follows by combining the convex combination property of distributional Retrace in Lemma 3.1 with the $p$-convexity of $L_p$ distance [10]. □

### E.1  Categorical representation

In categorical representations [23], we consider parametric distributions of the form for a fixed $m \geq 1$, $\sum_{i=1}^m p_i \delta_{z_i}$, where $(z_i)_{i=1}^m \in \mathbb{R}$ are a fixed set of atoms and $(p_i)_{i=1}^m$ is a categorical distribution such that $\sum_{i=1}^m p_i = 1$ and $p_i \geq 0$. Denote the class of such distributions as $\mathscr{P}_\mathcal{C}(\mathbb{R}) := \{\sum_{i=1}^m p_i \delta_{z_i} | \sum_{i=1}^m p_i = 1, p_i \geq 0\}$. For simplicity, we assume that the target return is supported on the set of atoms $[R_{\text{MIN}}/(1 - \gamma), R_{\text{MAX}}/(1 - \gamma)] \subset [z_1, z_m]$.

We introduce the projection that maps from an initial back-up distribution to the categorical parametric class: $\Pi_\mathcal{C} : \mathscr{P}_\infty(\mathbb{R}) \to \mathscr{P}_\mathcal{C}(\mathbb{R})$ defined as $\Pi_\mathcal{C}\eta := \arg\min_{\nu \in \mathscr{P}_\mathcal{C}(\mathbb{R})} L_2(\nu, \eta), \forall \nu \in \mathscr{P}_\infty(\mathbb{R})$. The projection can be easily calculated as described in [6, 17]. For any distribution vector $\eta \in \mathscr{P}_\infty(\mathbb{R})^{\mathcal{X} \times \mathcal{A}}$, define $\Pi_\mathcal{C}\eta$ as the component-wise projection. Now, given the composed operator $\Pi_\mathcal{C}\mathcal{R}^{\pi,\mu} : \mathscr{P}_\infty(\mathbb{R})^{\mathcal{X} \times \mathcal{A}} \to \mathscr{P}_\mathcal{C}(\mathbb{R})^{\mathcal{X} \times \mathcal{A}}$, we characterize the convergence of the sequence $\eta_k = (\Pi_\mathcal{C}\mathcal{R}^{\pi,\mu})^k \eta_0$.

**Theorem E.2.** (**Convergence of categorical distributions**) The projected distributional Retrace operator $\Pi_\mathcal{C}\mathcal{R}^{\pi,\mu}$ is $\beta_{L_2}$-contractive under $\overline{L}_2$ distance in $\mathscr{P}_\mathcal{Q}(\mathbb{R})$. As a result, the above $\eta_k$ converges to a limiting distribution $\eta_\mathcal{R}^\pi$ in $\overline{L}_2$, such that $\overline{L}_2(\eta_k, \eta_\mathcal{R}^\pi) \leq (\beta_{L_2})^k \overline{L}_2(\eta_0, \eta_\mathcal{R}^\pi)$. Further, the quality of the fixed point is characterized as $\overline{L}_2(\eta_\mathcal{R}^\pi, \eta^\pi) \leq (1 - \beta_{L_2})^{-1} \overline{L}_2(\Pi_\mathcal{C}\eta^\pi, \eta^\pi)$.

*Proof.* The above theorem follows from Lemma E.1. Indeed, since $\Pi_\mathcal{Q}$ is a non-expansion in supremum Cramér distance $\overline{L}_2$ [17], the composed operator $\Pi_\mathcal{Q}\mathcal{R}^{\pi,\mu}$ is $\beta_{L_2}$-contractive in $\overline{L}_2$. Following the same argument as the proof of Theorem 5.1, we obtain the remaining desired results. □

The distributional Retrace operator also improves over one-step distributional Bellman operator in two aspects: (1) the bound on the contraction rate $\beta_{L_2} \leq \sqrt{\gamma}$ is smaller, usually leading to faster contraction to the fixed point; (2) the bound on the quality of the fixed point is improved.

### E.2  Cross-entropy update and C51-Retrace

Unlike in the quantile projection case, where calculating $\Pi_\mathcal{Q}\eta$ requires solving a quantile regression minimization problem, the categorical projection can be calculated in an analytic way [17, 10]. Assume the categorical distribution is parameterized as $\eta_w(x, a) = \sum_{i=1}^m p_i(x, a; w) \delta_{z_i}$. After

computing the back-up target distribution $\Pi_{\mathcal{C}}\mathcal{R}^{\pi,\mu}\eta(x,a)$ for a given distribution vector $\eta$, the algorithm carries out a gradient-based incremental update

$$w \leftarrow w - \alpha \nabla_w \mathbb{CE}\left[\Pi_{\mathcal{C}}\mathcal{R}^{\pi,\mu}\eta(x,a)|\eta_w(x,a)\right],$$

where $\mathbb{CE}(p|q) \coloneqq -\sum_i p_i \log q_i$ denotes the cross-entropy between distribution $p$ and $q$. For simplicity, we adopt a short-hand notation $\mathbb{CE}(\eta|\eta_w) = \mathbb{CE}_w(\eta)$. Note also that in practice, $\eta$ can be a slowly updated copy of $\eta_w$ [33]. As such, the gradient-based update can be understood as approximating the iteration $\eta_{k+1} = \mathcal{R}^{\pi,\mu}\eta_k$. We propose the following unbiased estimate to the cross-entropy $\widehat{\mathbb{CE}}_w\left[\Pi_{\mathcal{C}}\mathcal{R}^{\pi,\mu}\eta(x,a)\right]$, calculated as follows

$$\mathbb{CE}_w(\eta(x,a)) + \sum_{t=0}^{\infty} c_{1:t}\left(\mathbb{CE}_w\left((\mathsf{b}_{t+1})_{\#}\,\eta\left(X_{t+1},A_{t+1}^{\pi}\right)\right) - \mathbb{CE}_w\left((\mathsf{b}_t)_{\#}\,\eta(X_t,A_t)\right)\right).$$

**Lemma E.3.** (**Unbiased stochastic estimate for categorical update**) Assume that the trajectory terminates within $H < \infty$ steps almost surely, then we have $\mathbb{E}_\mu\left[\widehat{\mathbb{CE}}_w\left(\Pi_{\mathcal{C}}\mathcal{R}^{\pi,\mu}\eta(x,a)\right)\right] = \mathbb{CE}_w\left(\Pi_{\mathcal{C}}\mathcal{R}^{\pi,\mu}\eta(x,a)\right)$. Without loss of generality, assume $w$ is a scalar parameter. If there exists a constant $M > 0$ such that $|\nabla_w \mathbb{CE}_w(\eta)| \leq M, \forall \eta \in \mathscr{P}_\infty(\mathbb{R})$, then the gradient estimate is also unbiased $\mathbb{E}_\mu\left[\nabla_w \widehat{\mathbb{CE}}_w\left(\Pi_{\mathcal{C}}\mathcal{R}^{\pi,\mu}\eta(x,a)\right)\right] = \nabla_w \mathbb{CE}_w\left(\Pi_{\mathcal{C}}\mathcal{R}^{\pi,\mu}\eta(x,a)\right)$.

*Proof.* The cross-entropy is defined for any distribution $\mathbb{CE}_w(\eta)$. For any signed measure $\nu = \sum_{i=1}^m w_i\eta_i$ with $\eta_i \in \mathscr{P}_\infty(\mathbb{R})$, we define the generalized cross-entropy as

$$\mathbb{CE}_w(\nu) \coloneqq \sum_{i=1}^m w_i \mathbb{CE}_w(\eta_i),$$

Next, we note the cross-entropy is linear in the input distribution (or signed measure). In particular, for a set of $N$ (potentially infinite) coefficients and distributions (signed measures) $(a_i, \eta_i)$,

$$\mathbb{CE}_w\left(\sum_{i=1}^N a_i\eta_i\right) \coloneqq \sum_{i=1}^m a_i \mathbb{CE}_w(\eta_i).$$

When $a_i$ denotes a distribution, the above rewrites as $\mathbb{CE}_w(\mathbb{E}[\eta_i]) = \mathbb{E}[\mathbb{CE}(\eta_i)]$. Finally, combining everything together, we have $\mathbb{E}_\mu\left[\widehat{\mathbb{CE}}_w\left(\Pi_{\mathcal{C}}\mathcal{R}^{\pi,\mu}\eta(x,a)\right)\right]$ evaluate to

$$= \mathbb{E}_\mu\left[\mathbb{CE}_w(\eta(x,a)) + \sum_{t=0}^{\infty} c_{1:t}\left(\mathbb{CE}_w\left((\mathsf{b}_{t+1})_{\#}\,\eta\left(X_{t+1},A_{t+1}^{\pi}\right)\right) - \mathbb{CE}_w\left((\mathsf{b}_t)_{\#}\,\eta(X_t,A_t)\right)\right)\right]$$

$$=_{(a)} \mathbb{E}_\mu\left[\mathbb{CE}\left(\widehat{\mathcal{R}}^{\pi,\mu}\eta(x,a)\right)\right] =_{(b)} \mathbb{E}_\mu\left[\mathbb{CE}\left(\mathcal{R}^{\pi,\mu}\eta(x,a)\right)\right].$$

In the above, (a) follows from the definition of the cross-entropy with signed measure $\widehat{\mathcal{R}}^{\pi,\mu}\eta(x,a)$ and (b) follows from the linearity property of cross-entropy.

Next, to show that the gradient estimate is unbiased too, the high level idea is to apply dominated convergence theorem (DCT) to justify the exhchange of gradient and expectation [34]. This is similar to the quantile representation case (see proof for Lemma 5.2). To this end, consider the absolute value of the gradient estimate $\left|\nabla_w \widehat{\mathbb{CE}}_w\left(\mathcal{R}^{\pi,\mu}\eta(x,a)\right)\right|$, which serves as an upper bound to the gradient estimate. In order to apply DCT, we need to show the expectation of the absolute gradient is finite. Note we have

$$\mathbb{E}_\mu\left[\left|\nabla_w \widehat{\mathbb{CE}}_w\left(\mathcal{R}^{\pi,\mu}\eta(x,a)\right)\right|\right]$$

$$= \mathbb{E}_\mu\left[\left|\nabla_w \mathbb{CE}_w(\eta(x,a)) + \sum_{t=0}^H c_{1:t}\left(\nabla_w \mathbb{CE}_w\left((\mathsf{b}_{t+1})_{\#}\,\eta\left(X_{t+1},A_{t+1}^{\pi}\right)\right) - \nabla_w \mathbb{CE}_w\left((\mathsf{b}_t)_{\#}\,\eta(X_t,A_t)\right)\right)\right|\right]$$

$$\leq_{(a)} \mathbb{E}_\mu\left[|\nabla_w \mathbb{CE}_w(\eta(x,a))| + \sum_{t=0}^H c_{1:t}\left|\nabla_w \mathbb{CE}_w\left((\mathsf{b}_{t+1})_{\#}\,\eta\left(X_{t+1},A_{t+1}^{\pi}\right)\right) - \nabla_w \mathbb{CE}_w\left((\mathsf{b}_t)_{\#}\,\eta(X_t,A_t)\right)\right|\right]$$

$$\leq_{(b)} \mathbb{E}_\mu\left[M + \sum_{t=0}^H \rho^t \cdot M\right] < \infty,$$

where (a) follows from the application of triangle inequality; (b) follows from the fact that the QR loss gradient against a fixed distribution is bounded $\nabla_w \mathbb{CE}_w(\nu) \in [-M, M], \forall \nu \in \mathscr{P}_\infty(\mathbb{R})$ [16].

Hence, with the application DCT, we can exchange the gradient and expectation operator, which yields $\mathbb{E}_\mu \left[ \nabla_w \widehat{\mathbb{CE}}_w^\tau (\mathcal{R}^{\pi,\mu} \eta(x,a)) \right] = \nabla_w \mathbb{E}_\mu \left[ \widehat{\mathbb{CE}}_w^\tau (\mathcal{R}^{\pi,\mu} \eta(x,a)) \right] = \nabla_w \mathbb{CE}_w (\mathcal{R}^{\pi,\mu} \eta(x,a))$.

$\square$

We remark that the condition on the bounded gradient $|\nabla_w \mathbb{CE}_w(\eta)| \leq M$ is not restrictive. When $\eta_w$ is adopts a softmax parameterization and $w$ represents the logits, $M = 1$.

Finally, the deep RL agent C51 parameterizes the categorical distribution $p_i(x, a; w)$ with a neural network $w$ at each state action pair $(x, a)$ [23]. When combined with the above algorithm, this produces C51-Retrace.

# F  Additional experiment details

In this section, we provide detailed information about experiment setups and additional results. All experiments are carried out in Python, using NumPy for numerical computations [35] and Matplotlib for visualization [36]. All deep RL experiments are carried out with Jax [37], specifically making use of the DeepMind Jax ecosystem [38].

## F.1  Tabular

We provide additional details on the tabular RL experiments.

**Setup.** We consider a tabular MDP with $|\mathcal{X}| = 3$ states and $|\mathcal{A}| = 2$ actions. The reward $r(x, a)$ is deterministic and generated from a standard Gaussian distribution. The transition probability $P(\cdot|x, a)$ is sampled from a Dirichlet distribution with parameter $(\Gamma, \Gamma...\Gamma)$ for $\Gamma = 0.5$. The discount factor is fixed as $\gamma = 0.9$. The MDP has a starting state-action pair $(x_0, a_0)$. The behavior policy $\mu$ is a uniform policy. The target policy is generated as follows: we first sample a deterministic policy $\pi_d$ and then compute $\pi = (1 - \varepsilon)\pi_d + \varepsilon\mu$, with parameter $\varepsilon$ to control the level of off-policyness.

**Quantile distribution and projection.** We use $m = 100$ atoms throughout the experiments. Assuming access to the MDP parameters (e.g., reward and transition probability), we can analytically compute the projection $\Pi_\mathcal{Q}$ using a sorting algorithm. See [16, 10] for details.

**Evaluation metrics.** Let $\eta_k = (\mathcal{R}^{\pi,\mu})^k \eta_0$ be the $k$-th iterate. We use a few different metrics in Figure 3. Given any particular distributional Retrace operator $\mathcal{R}^{\pi,\mu}$, there exists a fixed point to the composed operator $\Pi_\mathcal{Q} \mathcal{R}^{\pi,\mu}$. Recall that we denote this distribution as $\eta_\mathcal{R}^\pi$. Fig 3(a)-(b) calculates the iterates' distance from the fixed point, evaluated at $(x_0, a_0)$.

$$L_p \left( \eta_k(x_0, a_0), \eta_\mathcal{R}^\pi(x_0, a_0) \right).$$

Fig 3(c) calculates the distance from the projected target distribution $\Pi_\mathcal{Q} \eta^\pi$. Recall that $\Pi_\mathcal{Q} \eta^\pi$ is in some sense the best possible approximation that the current quantile representation can obtain.

$$L_p \left( \eta_k(x_0, a_0), \Pi_\mathcal{Q} \eta^\pi(x_0, a_0) \right).$$

## F.2  Deep reinforcement learning

We provide additional details on the deep RL experiments.

**Evaluation metrics.** For the $i$-th of the 57 Atari games, we obtain the performance of the agent $G_i$ at any given point in training. The normalized performance is computed as $Z_i = (G_i - U_i)/(H_i - U_i)$ where $H_i$ is the human performance and $U_i$ is the performance of a random policy. Then the mean/median metric is calculated as the mean or median statistics over $(Z_i)_{i=1}^{57}$.

The super human ratio is computed as the number of games such as $Z_i \geq 1$, i.e., $G_i \geq H_i$ where the agent obtains super human performance on the game. Formally, it is compute as $\frac{1}{57} \sum_{i=1}^{57} \mathbb{I}[Z_i \geq 1]$.

**Shared properties of all baseline agents.** All baseline agents use the same torso architecture as DQN [33] and differ in the head outputs, which we specify below. All agents an Adam optimizer [39] with a fixed learning rate; the optimization is carried out on mini-batches of size 32 uniformly sampled from the replay buffer. For exploration, the agent acts $\varepsilon$-greedy with respect to induced Q-functions, the details of which we specify below. The exploration policy adopts $\varepsilon$ that starts with $\varepsilon_{\max} = 1$ and linearly decays to $\varepsilon_{\min} = 0.01$ over training. At evaluation time, the agent adopts $\varepsilon = 0.001$; the small exploration probability is to prevent the agent from getting stuck.

**Details of baseline C51 agent.** The agent head outputs a matrix of size $|\mathcal{A}| \times m$, which represents the logits to $(p_i(x, a; \theta))_{i=1}^m$. The support $(z_i)_{i=1}^m$ is generated as a uniform array over $[-V_{\mathrm{MAX}}, V_{\mathrm{MAX}}]$. Though $V_{\mathrm{MAX}}$ should in theory be determined by $R_{\mathrm{MAX}}$; in practice, it has been found that setting $V_{\mathrm{MAX}} = R_{\mathrm{MAX}}/(1 - \gamma)$ leads to highly sub-optimal performance. This is potentially because usually the random returns are far from the extreme values $R_{\mathrm{MAX}}/(1 - \gamma)$, and it is better to set $V_{\mathrm{MAX}}$ at a smaller value. Here, we set $V_{\mathrm{MAX}} = 10$ and $m = 51$. For details of other hyperparameters, see [6]. The induced Q-function is computed as $Q_\theta(x, a) = \sum_{i=1}^m p_i(x, a; \theta) z_i$.

**Details of baseline QR-DQN agent.** The agent head outputs a matrix of size $|\mathcal{A}| \times m$, which represents the quantile locations $(z_i(x, a; \theta))_{i=1}^m$. Here, we set $m = 201$. For details of other hyperparameters, see [16]. The induced Q-function is computed as $Q_\theta(x, a) = \frac{1}{m} \sum_{i=1}^m z_i(x, a; \theta)$.

**Details of multi-step agents.** Multi-step variants use exactly the same hyperparameters as the one-step baseline agent. The only difference is that the agent uses multi-step back-up targets.

The agent stores partial trajectories $(X_t, A_t, R_t, x_t)_{t=0}^{n-1} \sim \mu$ generated under the behavior policy. Here, the behavior policy $\mu$ is the $\varepsilon$-greedy policy with respect to a potentially old Q-function (this is because the data at training time is sampled from the replay); the target policy $\pi$ is the greedy policy with respect to the current Q-function.

# G Proof

To simplify the proof, we assume that the immediate random reward takes a finite number of values. It is straightforward to generalize results to the case where the reward takes an infinite number of values (e.g., the random reward has a continuous distribution).

**Assumption G.1. (Reward takes a finite number of values)** For all state-action pair $(x, a)$, we assume the random reward $R(x, a)$ takes a finite number of values. Let $\widetilde{R}$ be the finite set of values that the reward $\{R(x, a), (x, a) \in \mathcal{X} \times \mathcal{A}\}$ can take.

For any integer $t \geq 1$, Let $\widetilde{R}^t$ denotes the Cartesian product of $t$ copies of $\widetilde{R}$:

$$\widetilde{R}^t := \underbrace{\widetilde{R} \times \widetilde{R} \times ... \times \widetilde{R}}_{t \text{ copies of } \widetilde{R}}.$$

For any fixed $t$, we let $r_{0:t-1}$ denote the sequence of realizable rewards from time 0 to time $t - 1$. Since $\widetilde{R}$ is a finite set, $\widetilde{R}^t$ is also a finite set.

**Lemma 3.1. (Convex combination)** The Retrace back-up target is a convex combination of $n$-step target distributions. Formally, there exists an index set $I(x, a)$ such that $\mathcal{R}^{\pi,\mu}\eta(x, a) = \sum_{i \in I(x,a)} w_i \eta_i$ where $w_i \geq 0$, $\sum_{i \in I(x,a)} w_i = 1$ and $(\eta_i)_{i \in I(x,a)}$ are $n_i$-return target distributions.

*Proof.* In general $c_t = c(F_t, A_t)$ where $F_t$ is a filtration of $(X_s, A_s)_{s=0}^t$. To start with, we assume $c_t = c(X_t, A_t)$ to be a Markovian trace coefficient [13]. We start with the simpler case because the proof is greatly simplified with notations and can extend to the general case with some care. We discuss the extension to the general case where $c_t = c(F_t, A_t)$ towards the end of the proof.

For all $t \geq 1$, we define the coefficient

$$w_{y,b,r_{0:t-1}} := \mathbb{E}_\mu \left[ c_1 ... c_{t-1} \left( \pi(b|X_t) - c(X_t, b)\mu(b|X_t) \right) \cdot \mathbb{I}[X_t = y] \Pi_{s=0}^{t-1} \mathbb{I}[R_s = r_s] \right].$$

Through careful algebra, we can rewrite the Retrace operator as follows

$$\mathcal{R}^{\pi,\mu}\eta(x, a) = \sum_{t=1}^\infty \sum_{y \in \mathcal{X}} \sum_{b \in \mathcal{A}} \sum_{r_{0:t-1} \in \widetilde{R}^t} w_{y,b,r_{0:t-1}} \left( \mathsf{b}_{G_{0:t-1}, \gamma^t} \right)_\# \eta(y, b).$$

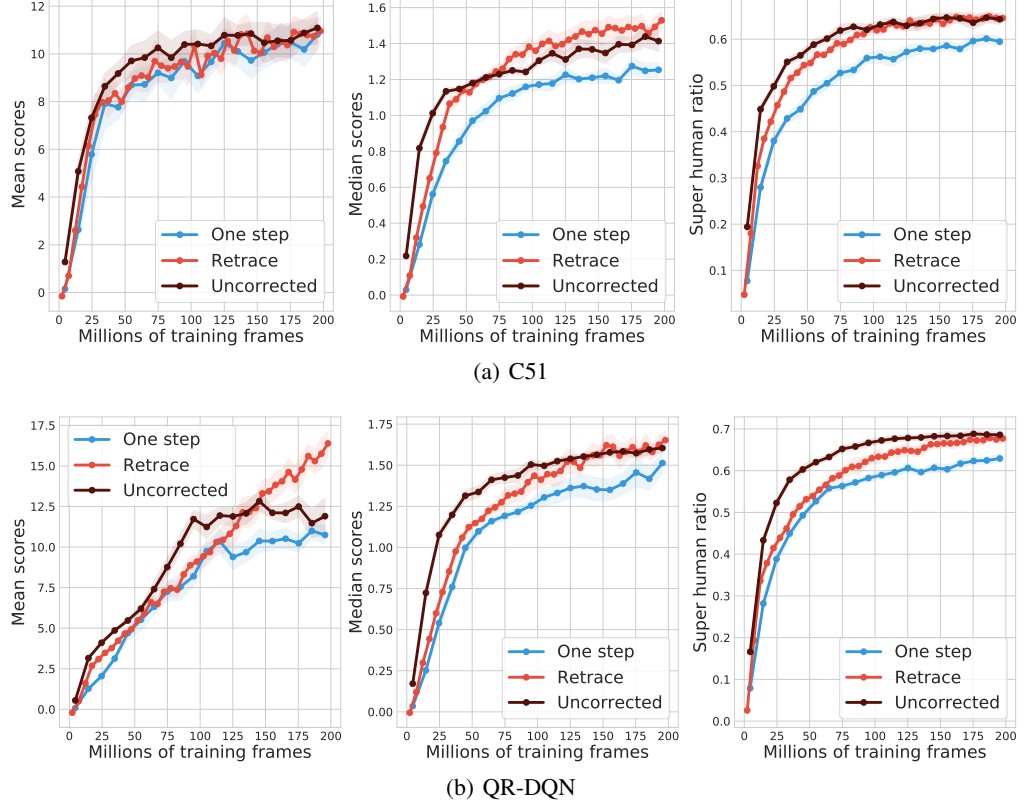

(a) C51

(b) QR-DQN

Figure 6: Deep RL experiments on Atari-57 games for (a) C51 and (b) QR-DQN. We compare the one-step baseline agent against the multi-step variants (Retrace and uncorrected $n$-step). For all multi-step variants, we use $n = 3$. For each agent, we calculate the mean, median and super human ratio performance across all games, and we plot the mean $\pm$ standard error across 3 seeds. In almost all settings, Multi-step variants provide clear advantage over the one-step baseline algorithm.

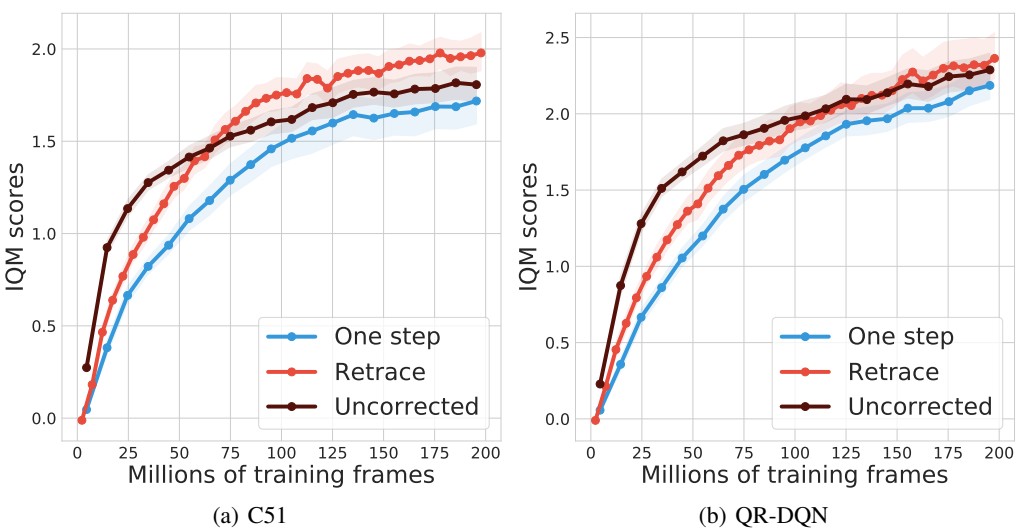

(a) C51

(b) QR-DQN

Figure 7: Deep RL experiments on Atari-57 games for (a) C51 and (b) QR-DQN, with the same setup as in Figure 7. Here, we compute the interquartile mean (IQM) with 95% bootstrapped confidence interval [40]. In a nutshell, IQM calculates the mean scores after removing extreme score values, making the performance statistics more robust. Even after excluding extreme scores, Retrace obtains favorable performance compared to the uncorrected and one-step algorithm.

Note that each term of the form $\left(\mathsf{b}_{G_{0:t-1},\gamma^t}\right)_{\#}\eta(y,b)$ corresponds to applying a pushforward operation $\left(\mathsf{b}_{G_{0:t-1},\gamma^t}\right)_{\#}$ on the distribution $\eta(x,a)$, which means $\left(\mathsf{b}_{G_{0:t-1},\gamma^t}\right)_{\#}\eta(y,b) \in \mathscr{P}_{\infty}(\mathbb{R})$. Now that we have expressed $\mathcal{R}^{\pi,\mu}\eta(x,a)$ as a linear combination of distributions, we proceed to show that the combination is in fact convex.

Under the assumption $c_t \in [0,\rho_t]$, we have $\pi(b|y) - c(y,b)\mu(b|y) \geq 0$ for all $(y,b) \in \mathcal{X} \times \mathcal{A}$. Therefore, all weights are non-negative. Next, we examine the sum of all coefficients $\sum w_{y,b,r_{0:t-1}} = \sum_{t=1}^{\infty} \sum_{x\in\mathcal{X}} \sum_{b\in\mathcal{A}} \sum_{r_{0:t-1}\in\widetilde{R}^t} w_{y,b,r_{0:t-1}}$.

$$\sum w_{y,b,r_{0:t-1}} =_{(a)} \sum_{t=1}^{\infty} \sum_{y\in\mathcal{X}} \sum_{b\in\mathcal{A}} \mathbb{E}_{\mu}\left[c_1...c_{t-1}\left(\pi(b|X_t) - c(X_t,b)\mu(b|X_t)\right) \cdot \mathbb{I}[X_t = y]\right]$$

$$=_{(b)} \sum_{t=1}^{\infty} \mathbb{E}_{\mu}\left[c_1...c_{t-1}(1-c_t)\right] =_{(c)} 1.$$

In the above, (a) follows from the fact that $\sum_{r_s\in\widetilde{R}} \mathbb{E}[\mathbb{I}[R_s = r_s]] = 1$; (b) follows from the fact that for all time steps $t \geq 1$, the following is true,

$$\sum_{y\in\mathcal{X}} \sum_{b\in\mathcal{A}} \mathbb{E}_{\mu}\left[c_1...c_{t-1}\left(\pi(b|X_t) - c(X_t,b)\mu(b|X_t)\right) \cdot \mathbb{I}[X_t = y]\right]$$

$$= \sum_{b\in\mathcal{A}} \mathbb{E}_{\mu}\left[c_1...c_{t-1}\left(\pi(b|X_t) - c(X_t,b)\mu(b|X_t)\right)\right]$$

$$= \mathbb{E}_{\mu}\left[c_1...c_{t-1}\left(1 - \sum_{b\in\mathcal{A}} c(X_t,b)\mu(b|X_t)\right)\right]$$

$$= \mathbb{E}_{\mu}\left[c_1...c_{t-1}(1-c_t)\right].$$

Finally, (c) is based on the observation that the summation telescopes. Now, by taking the index set to be the set of indices that parameterize $w_{y,b,r_{0:t-1}}$,

$$I(x,a) = \cup_{t=1}^{\infty} (y,b,r_{0:t-1})_{y\in\mathcal{X},b\in\mathcal{A},r_{0:t-1}\in\widetilde{R}^t}.$$

We can write $\mathcal{R}^{\pi,\mu}\eta(x,a) = \sum_{i\in I(x,a)} w_i\eta_i$. Note further that for any $i \in I(x,a)$, $\eta_i = \left(\mathsf{b}_{G_{0:t-1},\gamma^t}\right)_{\#}\eta(y,b)$ is a fixed distribution. The above result suggests that $\mathcal{R}^{\pi,\mu}\eta(x,a)$ is a convex combination of fixed distributions.

**Extension to the general case.** When $c_t = c(F_t, A_t)$ is filtration dependent, we let $\mathcal{F}_t$ to be the space of the filtration value up to time $t$. For simplicity with the notation, we assume $\mathcal{F}_t$ contains a finite number of elements, such that below we can adopt the summation notation instead of integral. Define the combination coefficient

$$w_{y,b,f_t,r_{0:t-1}} := \mathbb{E}_{\mu}\left[c_1...c_{t-1}\left(\pi(b|X_t) - c(F_t,b)\mu(b|X_t)\right) \cdot \mathbb{I}[X_t = y]\Pi_{s=0}^{t-1}\mathbb{I}[R_s = r_s]\right].$$

It is straightforward to verify the following

$$\mathcal{R}^{\pi,\mu}\eta(x,a) = \sum_{t=1}^{\infty} \sum_{y\in\mathcal{X}} \sum_{b\in\mathcal{A}} \sum_{f_t\in\mathcal{F}_t} \sum_{r_{0:t-1}\in\widetilde{R}^t} w_{y,b,f_t,r_{0:t-1}} \left(\mathsf{b}_{G_{0:t-1},\gamma^t}\right)_{\#}\eta(y,b).$$

In addition, the combination coefficients $w_{y,b,f_t,r_{0:t-1}}$ sum to 1 and are all non-negative. $\square$

**Proposition 3.2.** (**Contraction**) $\mathcal{R}^{\pi,\mu}$ is $\beta$-contractive under supremum $p$-Wasserstein distance, where $\beta = \max_{x\in\mathcal{X},a\in\mathcal{A}} \sum_{t=1}^{\infty} \mathbb{E}_{\mu}\left[c_1...c_{t-1}(1-c_t)\right]\gamma^t \leq \gamma$.

*Proof.* From the proof of Lemma 3.1, we have

$$\mathcal{R}^{\pi,\mu}\eta(x,a) = \sum_{t=1}^{\infty} \sum_{y\in\mathcal{X}} \sum_{b\in\mathcal{A}} \sum_{r_{0:t-1}\in\widetilde{R}^t} w_{y,b,r_{0:t-1}} \left(\mathsf{b}_{G_{0:t-1},\gamma^t}\right)_{\#}\eta(y,b).$$

Now, we have for any $\eta_1, \eta_2 \in \mathscr{P}_\infty(\mathbb{R})^{\mathcal{X} \times \mathcal{A}}$, for any fixed $(x, a)$, we have $W_p\left(\mathcal{R}^{\pi,\mu}\eta_1(x, a), \mathcal{R}^{\pi,\mu}\eta_2(x, a)\right)$ upper bounded as follows

$$\leq_{(a)} \sum_{t=1}^{\infty} \sum_{y \in \mathcal{X}} \sum_{b \in \mathcal{A}} w_{y,b,r_{0:t-1}} W_p\left(\left(\mathsf{b}_{\sum_{s=0}^{t-1} \gamma^s r_s, \gamma^t}\right)_{\#} \eta_1(y, b), \left(\mathsf{b}_{\sum_{s=0}^{t-1} \gamma^s r_s, \gamma^t}\right)_{\#} \eta_2(y, b)\right)$$

$$\leq_{(b)} \sum_{t=1}^{\infty} \sum_{y \in \mathcal{X}} \sum_{b \in \mathcal{A}} w_{y,b,r_{0:t-1}} \gamma^t W_p\left(\eta_1(y, b), \eta_2(y, b)\right)$$

$$\leq_{(c)} \sum_{t=1}^{\infty} \sum_{y \in \mathcal{X}} \sum_{b \in \mathcal{A}} w_{y,b,r_{0:t-1}} \gamma^t \overline{W}_p\left(\eta_1, \eta_2\right)$$

In the above, (a) follows by applying the convexity of the $p$-Wasserstein distance [10]; (b) follows by the contraction property of the pushforward operation and $W_p$ [10]; (c) follows from the definition of $\overline{W}_p$. By taking the maixmum over $(x, a)$ on both sides of the inequality, we obtain

$$\overline{W}_p(\mathcal{R}^{\pi,\mu}\eta_1, \mathcal{R}^{\pi,\mu}\eta_2) \leq \beta \overline{W}_p(\eta_1, \eta_2).$$

This concludes the proof. $\qquad \square$

**Lemma G.2.** For any fixed $(x, a)$ and scalar $c \in \mathbb{R}$,

$$(\mathsf{b}_{c,1})_{\#} \eta^\pi(x, a) = \mathbb{E}_\pi\left[(\mathsf{b}_{c+R_0, \gamma})_{\#} \eta^\pi(X_1, A_1) \mid X_0 = x, A_0 = a\right]. \tag{6}$$

*Proof.* Let $B_y := \{x < y | x \in \mathbb{R}\}$ be a subset of $\mathbb{R}$ indexed by $y \in \mathbb{R}$. Since the set of all such sets $\{B_y, y \in \mathbb{R}\}$ is dense in the sigma-field of $\mathbb{R}$ [34], if we can show for two measures $\eta_1, \eta_2$

$$\eta_1(B_y) = \eta_2(B_y), \forall y$$

then, $\eta_1(B) = \eta_2(B)$ for all Borel sets in $\mathbb{R}$. Hence, in the following, we seek to show

$$\left((\mathsf{b}_{c,1})_{\#} \eta^\pi(x, a)\right) B_y = \left(\mathbb{E}_\pi\left[(\mathsf{b}_{c+R_0, \gamma})_{\#} \eta^\pi(X_1, A_1)\right]\right) B_y, \forall y \in \mathbb{R} \tag{7}$$

Let $F^\pi(y; x, a) := P^\pi(G^\pi(x, a) \leq y) = \eta^\pi(x, a)(B_y), y \in \mathbb{R}$ be the CDF of random variable $G^\pi(x, a)$. The distributional Bellman equation in Equation (1) implies

$$F^\pi(y; x, a) = \mathbb{E}_\pi\left[F^\pi\left(\frac{y - R_0}{\gamma}; X_1, A_1\right)\right], \forall y \in \mathbb{R}.$$

For any constant $c \in \mathbb{R}$, let $y = y' - c$ and plug into the above equality,

$$F^\pi(y' - c; x, a) = \mathbb{E}_\pi\left[F^\pi\left(\frac{y' - c - R_0}{\gamma}; X_1, A_1\right)\right], \forall y' \in \mathbb{R}.$$

Note the LHS is $((\mathsf{b}_{c,1})_{\#} \eta^\pi(x, a))(B_y)$ while the RHS is $\left(\mathbb{E}_\pi\left[(\mathsf{b}_{c+R_0, \gamma})_{\#} \eta^\pi(X_1, A_1)\right]\right)(B_y)$. This implies that Equation (7) holds and we conclude the proof. $\qquad \square$

**Proposition 3.3.** (**Unique fixed point**) $\mathcal{R}^{\pi,\mu}$ has $\eta^\pi$ as the unique fixed point in $\mathscr{P}_\infty(\mathbb{R})^{\mathcal{X} \times \mathcal{A}}$.

*Proof.* To verify that $\eta^\pi$ is a fixed point, it is equivalent to show

$$\mathbb{E}_\mu\left[\sum_{t=0}^{n} c_{1:t}\left((\mathsf{b}_{G_{0:t}, \gamma^{t+1}})_{\#} \eta^\pi(X_{t+1}, A_{t+1}^\pi) - (\mathsf{b}_{G_{0:t-1}, \gamma^t})_{\#} \eta^\pi(X_t, A_t)\right)\right] = \mathbf{0}.$$

Here, the RHS term $\mathbf{0}$ denotes the zero measure, a measure such that for all Borel sets $B \subset \mathbb{R}$, $\mathbf{0}(B) = 0$. We now verify that each of the summation term is a zero measure, i.e.,

$$\mathbb{E}_\mu\left[c_{1:t}\left((\mathsf{b}_{G_{0:t}, \gamma^{t+1}})_{\#} \eta^\pi(X_{t+1}, A_{t+1}^\pi) - (\mathsf{b}_{G_{0:t-1}, \gamma^t})_{\#} \eta^\pi(X_t, A_t)\right)\right] = \mathbf{0}.$$

To see this, we follow the derivation below,

$$\mathbb{E}_\mu \left[ c_{1:t} \left( \left( \mathbf{b}_{G_{0:t},\gamma^{t+1}} \right)_\# \eta^\pi(X_{t+1}, A_{t+1}^\pi) - \left( \mathbf{b}_{G_{0:t-1},\gamma^t} \right)_\# \eta^\pi(X_t, A_t) \right) \right]$$

$$=_{(a)} \mathbb{E} \left[ \mathbb{E} \left[ c_{1:t} \left( \left( \left( \mathbf{b}_{G_{0:t},\gamma^{t+1}} \right)_\# \eta^\pi(X_{t+1}, A_{t+1}^\pi) \right) - \left( \mathbf{b}_{G_{0:t-1},\gamma^t} \right)_\# \eta^\pi(X_t, A_t) \right) \Big| (X_s, A_s, R_{s-1})_{s=1}^t \right] \right]$$

$$=_{(b)} \mathbb{E} \left[ c_{1:t} \mathbb{E} \left[ \left( \mathbf{b}_{G_{0:t},\gamma^{t+1}} \right)_\# \eta^\pi(X_{t+1}, A_{t+1}^\pi) \Big| (X_s, A_s, R_{s-1})_{s=1}^t \right] - c_{1:t} \left( \mathbf{b}_{G_{0:t-1},\gamma^t} \right)_\# \eta^\pi(X_t, A_t) \right]$$

$$=_{(c)} \mathbb{E} \left[ c_{1:t} \underbrace{\mathbb{E} \left[ \left( \mathbf{b}_{G_{0:t-1}+\gamma^t R_t,\gamma^{t+1}} \right)_\# \eta^\pi(X_{t+1}, A_{t+1}^\pi) \Big| (X_s, A_s, R_{s-1})_{s=1}^t \right]}_{\text{first term}} - c_{1:t} \left( \mathbf{b}_{G_{0:t-1},\gamma^t} \right)_\# \eta^\pi(X_t, A_t) \right].$$

$$(8)$$

In the above, in (a) we condition on $(X_s, A_s, R_s)_{s=1}^t$ and the equality follows from the tower property of expectations; in (b), we use the fact that the trace product $c_{1:t}$ and $\left( \mathbf{b}_{G_{0:t-1},\gamma^t} \right)_\# \eta^\pi(X_t, A_t)$ are deterministic function of the conditioning variable $(X_s, A_s, R_s)_{s=1}^t$; in (c), we split the summation $G_{0:t} = G_{0:t-1} + \gamma R_t$. Now we examine the first term in Equation (8), by applying Lemma G.2, we have

$$\text{first term} = \left( \mathbf{b}_{G_{0:t-1},\gamma^t} \right)_\# \eta^\pi(X_t, A_t).$$

This implies Equation (8) evaluates to a zero measure. Hence $\eta^\pi$ is a fixed point of the operator $\mathcal{R}^{\pi,\mu}$. Because $\mathcal{R}^{\pi,\mu}$ is also contractive by Proposition 3.2, the fixed point is unique. $\square$

**Theorem 5.1.** (**Convergence of quantile distributions**) The projected distributional Retrace operator $\Pi_\mathcal{Q}\mathcal{R}^{\pi,\mu}$ is $\beta$-contractive under $\overline{W}_\infty$ distance in $\mathscr{P}_\mathcal{Q}(\mathbb{R})$. As a result, the above $\eta_k$ converges to a limiting distribution $\eta_\mathcal{R}^\pi$ in $\overline{W}_\infty$, such that $\overline{W}_\infty(\eta_k, \eta_\mathcal{R}^\pi) \leq (\beta)^k \overline{W}_\infty(\eta_0, \eta_\mathcal{R}^\pi)$. Further, the quality of the fixed point is characterized as $\overline{W}_\infty(\eta_\mathcal{R}^\pi, \eta^\pi) \leq (1-\beta)^{-1} \overline{W}_\infty(\Pi_\mathcal{Q}\eta^\pi, \eta^\pi)$.

*Proof.* The quantile projection $\Pi_\mathcal{Q}$ is a non-expansion under $\overline{W}_\infty$ [16]. Since $\mathcal{R}^{\pi,\mu}$ is $\beta$-contractive under $\overline{W}_p$ for all $p \geq 1$, the composed operator $\Pi_\mathcal{Q}\mathcal{R}^{\pi,\mu}$ is $\beta$-contractive under $\overline{W}_\infty$. Now, because (1) $\Pi_\mathcal{Q}\mathcal{R}^{\pi,\mu} \in \mathscr{P}_\infty(\mathbb{R})^{\mathcal{X}\times\mathcal{A}}$; (2) the space $\Pi_\mathcal{Q}\mathcal{R}^{\pi,\mu} \in \mathscr{P}_\infty(\mathbb{R})^{\mathcal{X}\times\mathcal{A}}$ is closed [10]; (3) the operator is contractive, the iterate $\eta_k = (\Pi_\mathcal{Q}\mathcal{R}^{\pi,\mu})^k \eta_0$ converges to a limiting distribution $\eta_\mathcal{R}^\pi \in \mathscr{P}_\infty(\mathbb{R})^{\mathcal{X}\times\mathcal{A}}$. Finally, by Proposition 5.28 in [10], we have $\overline{W}_\infty(\eta_\mathcal{R}^\pi, \eta^\pi) \leq (1-\beta)^{-1} \overline{W}_\infty(\Pi_\mathcal{Q}\eta^\pi, \eta^\pi)$. $\square$

**Lemma 5.2.** (**Unbiased stochastic QR loss gradient estimate**) Assume that the trajectory terminates within $H < \infty$ steps almost surely, then we have $\mathbb{E}_\mu[\widehat{L}_{z_i(x,a)}^{\tau_i} (\mathcal{R}^{\pi,\mu}\eta(x,a))] = L_{z_i(x,a)}^{\tau_i} (\mathcal{R}^{\pi,\mu}\eta(x,a))$ and $\mathbb{E}_\mu[\nabla_{z_i(x,a)}\widehat{L}_{z_i(x,a)}^{\tau_i} (\mathcal{R}^{\pi,\mu}\eta(x,a))] = \nabla_{z_i(x,a)} L_{z_i(x,a)}^{\tau_i} (\mathcal{R}^{\pi,\mu}\eta(x,a))$.

*Proof.* The QR loss $L_\theta^\tau(\eta)$ is defined for any distribution $\eta$ and scalar parameter $\theta$. Let $\nu = \sum_{i=1}^m w_i \eta_i$ be the linear combination of distributions $(\eta_i)_{i=1}^m$ where $w_i$s are potentially negative coefficients. In this case, $\nu$ is a signed measure. We define the generalized QR loss for $\nu$ as the linear combination of QR losses against $\eta_i$ weighted by $w_i$,

$$L_\theta^\tau(\nu) := \sum_{i=1}^m w_i L_\theta^\tau(\eta_i).$$

Next, we note that the QR loss is linear in the input distribution (or signed measure). This means given any (potentially infinite) set of $N$ distributions or signed measures $\nu_i$ with coefficients $a_i$,

$$L_\theta^\tau \left( \sum_{i=1}^N a_i \nu_i \right) = \sum_{i=1}^N a_i L_\theta^\tau(\nu_i).$$

When $(a_i)_{i=1}^N$ denotes a distribution, the above is equivalently expressed as an exchange between expectation and the QR loss $L_\theta^\tau(\mathbb{E}[\nu_i]) = \mathbb{E}[L_\theta^\tau(\nu_i)]$. For notational convenience, we let $\theta = z_i(x, a)$ and $\tau = \tau_i$. Because the trajectory terminates within $H$ steps almost surely, since $c_{1:t} \leq \rho_{1:t} \leq \rho^H$

where $\rho := \max_{x \in \mathcal{X}, \mathcal{A}} \frac{\pi(a|x)}{\mu(a|x)}$, the estimate $\widehat{L}_\theta^\tau \left( \mathcal{R}^{\pi,\mu} \eta(x,a) \right)$ is finite almost surely. Combining all results from above we obtain the following

$$
\mathbb{E}_\mu \left[ \mathcal{R}^{\pi,\mu} \eta(x,a) \right] = \mathbb{E}_\mu \left[ L_\theta^\tau \left( \eta(x,a) \right) + \sum_{t=0}^{\infty} c_{1:t} \left( L_\theta^\tau \left( (\mathbf{b}_{t+1})_\# \eta \left( X_{t+1}, A_{t+1}^\pi \right) \right) - L_\theta^\tau \left( (\mathbf{b}_t)_\# \eta(X_t, A_t) \right) \right) \right]
$$

$$
=_{(a)} \mathbb{E}_\mu \left[ L_\theta^\tau \left( \widehat{\mathcal{R}}^{\pi,\mu} \eta(x,a) \right) \right] =_{(b)} L_\theta^\tau \left( \mathbb{E}_\mu \left[ \widehat{\mathcal{R}}^{\pi,\mu} \eta(x,a) \right] \right) = L_\theta^\tau \left( \mathcal{R}^{\pi,\mu} \eta(x,a) \right).
$$

In the above, (a) follows from the definition of the generalized QR loss against signed measure the definition of $\mathcal{R}^{\pi,\mu} \eta(x,a)$; (c) follows from the linearity of the QR loss.

Next, to show that the gradient estimate is unbiased too, the high level idea is to apply dominated convergence theorem (DCT) to justify the exhchange of gradient and expectation [34]. Since the expected QR loss gradient $\nabla_\theta L_\theta^\tau \left( \mathcal{R}^{\pi,\mu} \eta(x,a) \right)$ exists, we deduce that the estimate $\nabla_\theta \widehat{L}_\theta^\tau \left( \mathcal{R}^{\pi,\mu} \eta(x,a) \right)$ exists almost surely. Consider the absolute value of the gradient estimate $\left| \nabla_\theta \widehat{L}_\theta^\tau \left( \mathcal{R}^{\pi,\mu} \eta(x,a) \right) \right|$, which serves as an upper bound to the gradient estimate. In order to apply DCT, we need to show the expectation of the absolute gradient is finite. Note we have

$$
\mathbb{E}_\mu \left[ \left| \nabla_\theta \widehat{L}_\theta^\tau \left( \mathcal{R}^{\pi,\mu} \eta(x,a) \right) \right| \right]
$$

$$
= \mathbb{E}_\mu \left[ \left| \nabla_\theta L_\theta^\tau \left( \eta(x,a) \right) + \sum_{t=0}^{H} c_{1:t} \left( \nabla_\theta L_\theta^\tau \left( (\mathbf{b}_{t+1})_\# \eta \left( X_{t+1}, A_{t+1}^\pi \right) \right) - \nabla_\theta L_\theta^\tau \left( (\mathbf{b}_t)_\# \eta(X_t, A_t) \right) \right) \right| \right]
$$

$$
\leq_{(a)} \mathbb{E}_\mu \left[ \left| \nabla_\theta L_\theta^\tau \left( \eta(x,a) \right) \right| + \sum_{t=0}^{H} c_{1:t} \left| \nabla_\theta L_\theta^\tau \left( (\mathbf{b}_{t+1})_\# \eta \left( X_{t+1}, A_{t+1}^\pi \right) \right) - \nabla_\theta L_\theta^\tau \left( (\mathbf{b}_t)_\# \eta(X_t, A_t) \right) \right| \right]
$$

$$
\leq_{(b)} \mathbb{E}_\mu \left[ 1 + \sum_{t=0}^{H} \rho^t \cdot 2 \right] < \infty,
$$

where (a) follows from the application of triangle inequality; (b) follows from the fact that the QR loss gradient against a fixed distribution is bounded $\nabla_\theta L_\theta^\tau (\nu) \in [-1, 1], \forall \nu \in \mathscr{P}_\infty(\mathbb{R})$ [16].

With the application of DCT, we can exchange the gradient and expectation operator, which yields

$$
\mathbb{E}_\mu \left[ \nabla_\theta \widehat{L}_\theta^\tau \left( \mathcal{R}^{\pi,\mu} \eta(x,a) \right) \right] = \nabla_\theta \mathbb{E}_\mu \left[ \widehat{L}_\theta^\tau \left( \mathcal{R}^{\pi,\mu} \eta(x,a) \right) \right] = \nabla_\theta L_\theta^\tau \left( \mathcal{R}^{\pi,\mu} \eta(x,a) \right). \qquad \square
$$