# OpenReview forum: "The Nature of Temporal Difference Errors in Multi-step Distributional Reinforcement Learning"
_NeurIPS.cc/2022/Conference — NeurIPS 2022 Accept_

### Official Review · Reviewer_6egT · 2022-06-22

**Rating:** 5
**Confidence:** 5
**Soundness:** 3 good
**Presentation:** 3 good
**Contribution:** 3 good

**Summary:**

This paper investigates the multi-step off-policy distributional RL algorithms. Firstly, the multi-step distributional TD error is defined and the contraction property is also provided. Building this result, the authors pointed out the fundamental difference between multi-step distributional RL and classic RL by proposing a new notion called path-dependence distributional TD error. Finally, a new unbiased version of the multi-step distributional RL algorithm is proposed, including quantile-regression-Retrace and QR-DQN-Retrace, which is slightly better than the QRDQN and the naïve biased version of the multi-step distributional RL algorithm.

**Questions:**

1. Path-dependent distributional TD error. From Line 174 to 180, I feel that $G_{0:t-1}$ can still be canceled out in the definition of distributional TD error so that only R_t is left. I may miss some points, but could the authors provide more explanation about that? The more detailed derivation could help readers to have a better understanding of their discrepancy.

2. To demonstrate the key role of path-dependent distributional TD error, the authors only show that the path-independent operator numerically can be non-convergent. My suggestion is that a more rigorous proof of the non-convergent behavior could be more convincing.

3. The advantage of path-dependent distributional TD error needs to be further explored. As multi-step distributional TD may heavily depend on the past rewards, can it be beneficial for the optimization of RL algorithms? Or it can lead to an instability issue? A deeper explanation is welcome in the future version.

4. What is the motivation of the proposed unbiased QR loss in Line 280 and what is the advantage over the naïve biased version in [26]? It seems that the empirical improvement is not significant.

5. The experimental result across all Atari games can be also provided rather than only the mean and median of average rewards. For instance, authors can present a histogram to illustrate the improvement ratio of the proposed algorithm across all Atari games.


**Limitations:**

In the checklist, the authors said they described the limitations of their work. Maybe I missed them, but I have not found any discussion about them in the main context. This paper indeed makes some contribution to the multi-step off-policy distributional RL as claimed in the introduction, but perhaps the main limitation lies in that some results (both theoretically and empirically) are not inspiring or surprising enough that I have not learned too much after carefully reading this paper. For example, Proposition 3.2 may be a straightforward result based on previous results in the one-step case. Moreover, the underlying impact of the path-independent distributional TD error can be deeply investigated. The empirical improvement of the proposed unbiased Retrace algorithm seems to be only slightly better or even worse compared with the uncorrected version, as illustrated in Figure 6.


In summary, I faithfully thank the pain-taking efforts the authors make in order to make a solid contribution in the multi-step off-policy distributional RL setting, but the limitations may not be addressed easily. My suggestion is that the path-dependence notion can be further probed and the algorithm novelty can be improved to expect better performance. I would be happy to see a better version of this paper in the future.

**Strengths And Weaknesses:**

### Originality

Theoretically, the paper provides the first theoretical guarantees on multi-step off-policy distributional RL algorithms (see Proposition 3.2 and 3.3). To the best of my knowledge, the notion of path-dependent distributional TD error is novel. However, combining multi-step distributional RL with quantile regression is straightforward, although a relatively different unbiased version in Line 280 is derived.


### Quality

The writing quality is good. One minor typo in Line 239, “Speficially” should be “specifically”.

The technical contribution can be further improved. Generally, investigating the multi-step distributional RL may not be somewhat intriguing as some theoretical results in one-step distributional RL can normally be straightforward to extend into the multi-step version. As such, the theoretical result may not be surprising to me, and the design of the QR-based Retrace algorithm may not be novel enough.


###  Clarity

The paper is well organized and easy to follow. The key confusing point is the comparison between path-dependent distributional TD error and the value-based TD error, although I understand the authors did make some efforts to clarify it. Please refer to Question for my detailed concern.

###  Significance

One potential limitation of this paper is the proposed algorithm may not achieve significant improvement compared with the baseline “Uncorrected” as suggested in Figure 4. It is easy to expect that the multi-step distributional RL can be significantly better than the one-step and thus it is not surprising that “Retrace” and “Uncorrected” are superior to “one step”. However, as far as I can tell, the contribution in the algorithm part is the proposal of an unbiased version of QR loss (Line 280 and Lemma 5.2), but I can not recognize the improvement of the unbiased version in contrast to its biased version (“Uncorrected”).

---

> ### Author Response · Authors · 2022-08-02
> **Reply to your reviews**
>
> Thank you very much for your valuable inputs to our paper. We address your detailed comments below.
>
> ### **Question 1 on “path-dependent distributional TD error”**
>
> We include an in-depth discussion on the differences between path-dependent multi-step distributional TD errors and the path-independent alternatives in Appendix H. We provide an explanation on why the path-dependency cannot be removed and is fundamental to defining contractive operators. We also provide detailed motivations for path-independent TD errors, which are in fact more direct analogies to the value-based TD errors. We briefly explain why the path-dependency cannot be removed below.
>
> The cumulative sum of reward $G_{0:t-1}$ cannot be canceled out in the definition of multi-step distributional TD error. To see this more clearly, we can rewrite the multi-step distributional TD error as
>
> $$\tilde{\Delta}_{0:t}^\pi =  ( b_t ) \Delta_t^\pi$$
>
> where $b_t \coloneqq b_{G_{0:t-1},\gamma^t}$. In other words, the multi-step distributional TD error can be understood as applying a push-forward operation $b_t$ defined by $G_{0:t-1}$ to the one-step distributional TD error $\Delta_t^\pi$ at time t. Though the one-step distributional TD error depends on $R_t$ only, the push-forward in general depends on $G_{0:t-1}$ and does not cancel out $G_{0:t-1}$ as a common term.
>
> ### **Question 2 on “rigorous proof of non-convergent behavior”**
>
> Thanks for pointing this out. In Appendix H, we now provide a rigorous proof to show alternative path-independent operators (defined in line 192-193, line 203-206) are non-contractive in the space of signed measures with total mass 1, under the Cramer distance $L_p$ with $p=1$. Here, we give a brief sketch.
>
> The high-level idea is to construct a numerical counterexample on the same toy problem in the main paper: we can find two distributions $\eta_1$ and $\eta_2$ such that
>
> $$L_p( R \eta_1, R\eta_2) > L_p(\eta_1,\eta_2).$$
>
> Note that the proof technique of finding numerical counterexample is a common practice in RL theory, especially when proving that certain algorithms **do not** work. See, e.g., [1] shows the Boltzmann operator is numerically unstable on a toy problem; [2]  shows without off-policy linear TD is unstable on a toy problem.
>
> [1] Asadi et al., An alternative softmax operator for reinforcement learning, ICML 2017.
> [2] Sutton et al., An emphatic approach to the problem of off-policy temporal difference learning, JMLR 2016.
>
> ### **Question 3 on “advantage of path-dependent distributional TD error”**
>
> Thanks for your question on this. See **Appendix H** for a more comprehensive discussion.
>
> One major significance of path-dependent distributional TD error, is that path-dependency is indispensable in defining convergent multi-step off-policy operators. By drawing naive analogies to the value-based setting, one can plausibly come up with path-independent operators which are non-convergent (line 185 - line 206). As one of the examples we introduced in the paper (**Appendix C**), seeing that value-based operator takes the form
>
> $$RQ(x,a) = Q(x,a) + \mathbb{E}[\sum_{t=0}^\infty c_1...c_t \gamma^t \delta_t^\pi] $$
>
> Which can be seen as a weighted sum of one-step value-based TD error $\delta_t^\pi$, we can similarly weigh one-step distributional TD error $\Delta_t^\pi$ and construct a multi-step operator of the following form
>
> $$\bar{\mathcal{R}}\eta(x,a) = \eta(x,a) + \mathbb{E}[\sum_{t=0}^\infty c_1...c_t \gamma^t \Delta_t^\pi] $$
>
> As we showed numerically, such an operator is non-contractive (now we also have a formal theorem which states the operator is non-contractive in **Appendix H**). To build a theoretically grounded operator, we need to carefully probe the connection between n-step predictions and TD error in the distributional RL setting and exploit the fact that properly defined distributional TD errors need to be path-dependent.
>
> We will make sure to include more motivations and explanations in the revision.

---

> > ### Author Response · Authors · 2022-08-02
> > **Reply to your reviews (continued)**
> >
> >
> > ### **Question 4 on “motivation of unbiased QR loss”**
> >
> > To begin with, we want to note that extending an unbiased QR gradient estimate from the one-step case to the multi-step case is non-trivial. In showing Lemma 5.2, we have exploited a certain property of the QR loss function, namely its affineness in the target distribution. Such a result would not be generally available for other distributional loss functions.
> >
> > The uncorrected operator [26] does not apply off-policy corrections and hence differs significantly from the distributional Retrace operator we introduced in the paper. In tabular setting (Fig 3 (d)) we can verify that using an uncorrected operator leads to a biased fixed point, which is undesirable. In Atari games, the bias has been empirically shown to be more benign potentially because the behavior policy is sampled from a replay buffer which keeps getting updated with more on-policy data.
> >
> > Though it is true that the uncorrected operator can obtain most improvements over the one-step baseline, in the median performance of C51 and mean performance of QR-DQN (Fig 4), distributional Retrace still obtains some statistically significant improvements over the uncorrected operator. This hints at a favorable effect of carrying out off-policy corrections with such algorithms. Note that such improvements are not as visible in the value-based setting from prior literature.
> >
> > ### **Question 5 on “results across all atari games”**
> >
> > See **Appendix J** for per-game scores for the experiments in Fig 4.
> >
> > ### **Limitations of the paper**
> >
> > Thanks for mentioning this point. We have made some discussions in the conclusion section, in the form of potential future work and extension of the current paper. We will make sure to discuss limitations of the work more explicitly in the main paper.
> >
> > To be concrete, one open question is whether our method can be extended to more naive multi-step algorithms such as Q($\lambda$). For example, Q($\lambda$) [1] is a commonly-used off-policy learning algorithm but it does not satisfy the trace condition $c_t\leq\rho_t$ and therefore the theoretical statements in our work do not generally apply. It would be of interest to study the property of an equivalent “distributional Q($\lambda$)” algorithm.
> >
> > [1] Harutyunyan et al., Q($\lambda$) with off-policy corrections. ALT 2016.

---

> > > ### Comment · Reviewer_6egT · 2022-08-07
> > > **Thanks for the Reply.**
> > >
> > > Apologies to the authors for responding so late.
> > >
> > > I thank the authors for clarifying my questions. For Q1, 2, 3, and 5, after reading the response and appendix, I think most of my concerns have been clarified, although this further explanation should be placed in the main paper properly. There is also a typo in Lemma H.2 about the inequality order, which needs to be fixed.
> > >
> > > For Q4, I still feel the unbiased QR loss is not well motivated, although it is non-trivial. As the authors said, the biased loss may still be empirically useful for better exploration. Also, I find the proposed unbiased algorithm is not very empirically significant over the biased version at least for a lot of games, after checking the results of all Atari games that the authors provided.
> > >
> > > In summary, I am increasing my score to 5 vigilantly as I mainly appreciate the authors have clarified many of my concerns and the current paper version is well justified. Meanwhile, I do feel that this is still a borderline paper because there are still a couple of limitations to this work. Firstly, I am still not fully convinced about the key role and motivation of the path-dependency operator in the distributional RL setting. Secondly, the improvement of the unbiased QR-loss is not significant enough. Hope these suggestions could be helpful.

---

### Official Review · Reviewer_seSz · 2022-07-09

**Rating:** 7
**Confidence:** 4
**Soundness:** 3 good
**Presentation:** 3 good
**Contribution:** 3 good

**Summary:**

The authors study a multi-step off-policy learning approach to the distributional RL setting. The essential contributions of the paper may be seen as the theoretical and mathematical frameworks necessary to extend one-step distributional RL to an importance-sampled multi-step algorithm. This extension enables off-policy corrections for distributional RL in the same way that the Retrace algorithm enables off-policy corrections for value-based learning. Through a careful analysis, the authors elucidate the fact that such an extension of a Retrace style off-policy update to the distributional RL case is non-trivial. This fact is primarily owed to the nonlinearity, and repeated application of the pushforward operator used to produce the Bellman recurrence relation on the return distribution. The recurrence relation thus establishes a path-dependent relationship between the return distribution and the distributional TD error. This is in direct contrast to the path-independent relationship present in value-based RL methods.

The authors go on to substantiate the proposed methods theoretically and empirically. They evaluate their methods on Atari control problems and compare them against one-step distributional algorithms and uncorrected multi-step distributional algorithms. For each experiment, the authors run their algorithms using two different parametric distribution representations: categorical and quantile regressions. The proposed methods generally outperform the compared methods in either the mean or median score.

**Questions:**

-Could you please detail exactly in which way the pushforward operations are non-linear? I believe it to be non-obvious, considering that they are constructed using an affine measurable map, and both the composition and inverse of an affine map will be affine.

-Are there particular use cases or real-world consequences of the proposed approach that are worth mentioning? For example, are there specific applications that are now enabled by the work?


**Limitations:**

-The authors state in the checklist that they discuss the limitations of their work, although I could not find an explicit statement in the paper. This does not diminish the contributions of the paper.

**Strengths And Weaknesses:**

Strengths

-The work is, to the best of my knowledge, novel and significant, as it introduces a non-trivial extension of one-step distributional RL to the multi-step off-policy setting.

-The work provides a theoretical basis for the continued study of multi-step paradigms in off-policy reinforcement learning.

-The authors make clear that the extension provided in the paper deviates from the value-based multi-step paradigm.

-The paper provides adequate clarity of the introduced concepts and is precise with its mathematical notation.



Weaknesses

-The experimental results are not overly convincing that employing the multi-step distributional RL algorithm provides a significant empirical advantage.

-Minor grammatical mistake in line 357.

---

> ### Author Response · Authors · 2022-08-02
> **Reply to your revriews**
>
> Thank you very much for your valuable inputs to our paper. We address your detailed comments below.
>
> ### **Question 1 on “in which way the push-forward operations are nonlinear”**
>
> Thanks for your question, we will be sure to clarify more in the revision.
>
> To be clear, the push-forward operation is affine in the input distributions, in the sense that (we omit the hashtag # because it does not format well in the text box)
> $$\left(b_{r,\gamma}\right) (\alpha \nu_1 + (1-\alpha) \nu_2) = \alpha \left(b_{r,\gamma}\right) \nu_1 + (1-\alpha) \left(b_{r,\gamma} \right) \nu_2.$$
> However, the push-forward operation is not affine nor linear in the argument $r$ above. This is what we call the “nonlinearity” of the push-forward operations. We provide a bit more details below.
>
> More concretely, the nonlinearity of push-forward operations are mainly in contrast with the value-based counterparts. In Line 173-174, note that the value-based n-step predictions take the form $$G_{0:t-1} + \gamma^t Q_t.$$
>
> The prediction takes an “additive” form of $G_{0:t-1}$ and the bootstrapped target $Q_t$. The linearity of the addition operation ensures that we can cancel out $G_{0:t-1}$ as a common term from both of the n-step predictions. The distributional n-step predictions are computed as follows
>
> $$\left(b_{G_{0:t-1},\gamma^t}\right) \eta_t.$$
>
> The n-step prediction does not take e.g., an additive form as in the value-based setting (i.e., the above expression is not an additive function of $G_{0:t-1}$. This is the main source of nonlinearity we refer to in the paper. As a result, we do not expect in general $G_{0:t-1}$ to cancel out as a common term from the difference between two n-step predictions.
>
> ### **Question 2 on “particular use cases or real-world consequences”**
>
> Our proposal is a fairly generic multi-step distributional RL algorithm which can be applied to any applications where value-based learning or RL in general can be applied. The main difference might be that our approach could lead to potentially better asymptotic performance or entail faster learning. Such recent real world applications include [1], where they apply QR-DQN to train the agent.
>
> [1] Bellemare et al., Autonomous navigation of stratospheric balloons using RL, Nature 2020.
>
> ### **Limitations of the paper**
>
> Thanks for mentioning this point. We have made some discussions in the conclusion section, in the form of potential future work and extension of the current paper. We will make sure to discuss limitations of the work more explicitly in the main paper.
>
> To be concrete, one open question is whether our method can be extended to more naive multistep algorithms such as Q($\lambda$). For example, Q($\lambda$) [1] is a commonly used off-policy learning algorithm but it does not satisfy the trace condition $c_t\leq\rho_t$ and therefore the theoretical statements in our work do not generally apply. It would be of interest to study the property of an equivalent “distributional Q(lambda)” algorithm.
>
> [1] Harutyunyan et al., Q($\lambda$) with off-policy corrections. ALT 2016.

---

> > ### Comment · Reviewer_seSz · 2022-08-08
> > **Re: Response**
> >
> > Thank you for the response. I am still not convinced of the source of non-linearity in the pushforward operation. You mention that
> >
> > $(b_{G_{0:t-1}, \gamma^t})\eta$  is not an additive function of $G_{0:t-1}$, although I believe expanding this term with the definition of the pushforward operator in the paper (line no 76 in the revised edition) yields $G_{0:t-1} + \gamma^t\eta$, which is an additive function of $G_{0:t-1}$. Am I missing something?

---

> > > ### Author Response · Authors · 2022-08-08
> > > **Thank you for your response**
> > >
> > > We thank the reviewer for their response. We believe what the reviewer is pointing out is the difference between the transformation of the distribution itself, which is what our statement of non-linearity refers to, and the way this transformation applies to random variables.
> > >
> > > To confirm, the transformation of the distribution is non-linear as a function of $G_{0:t}$, although the corresponding transformation of random variables is affine in $G_{0:t}$. Importantly, it is the transformation of distributions that determines the algorithm behaviour, and this is reason that path-dependent TD errors are required, and why the operator becomes non-contractive if these are not used.
> > >
> > > In more detail, if we have a probability distribution $\nu$ (say the standard normal distribution, $N(0, 1)$), then the transformation $ (b_{g,1} )N(0, 1) $ is a new normal distribution, now with mean parameter $g$: $N(g, 1)$. This transformation can also be interpretted through a random variable distributed according to $\nu$. If $Z \sim \nu$, then $g + Z$ is distributed according to $(b_{g,1}) \nu$. It is true that the transformation of the random variable is **affine** in $g$ (a notion closely related to linearity), in the sense that for any $\lambda\in[0,1]$,
> > > $$( \lambda g_1 + (1-\lambda) g_2 ) + Z$$
> > > is the same as
> > > $$\lambda (g_1 + Z) + (1 - \lambda) (g_2 + Z).$$
> > > However, the transformation on the distributions themselves is not linear, nor affine, in $g$. This follows because
> > > $$(b_{\lambda g_1 + (1-\lambda) g_2, 1}) N(0, 1) = N(\lambda g_1 + (1 - \lambda) g_2, 1),$$
> > > but
> > > $$(\lambda b_{g_1, 1}) N(0, 1) + ((1-\lambda) b_{g_2, 1}) N(0, 1) = \lambda N(g_1, 1) + (1 - \lambda) N(g_2, 1),$$
> > > which is a mixture of Gaussians.
> > >
> > > Hope this clarifies, please let us know if you have other questions.

---

> > > > ### Comment · Reviewer_seSz · 2022-08-09
> > > > **All Clear**
> > > >
> > > > Thank you very much for the informative response, your argument is now clear to me! Could you please add a small note about this somewhere between line 175 and 180? Perhaps a footnote on 175 with part of your response above would suffice. Example below using your response:
> > > >
> > > > "Note: the transformation of the distribution is non-linear as a function of $G_{0:t}$, although the corresponding transformation of random variables is affine in $G_{0:t}$."
> > > >
> > > > To make this important distinction even more explicit, you could reproduce the exact example in your response above in an appendix in the paper.
> > > >
> > > > Thanks again!

---

> > > > > ### Author Response · Authors · 2022-08-09
> > > > > **Thank you for your reply**
> > > > >
> > > > > Thank you very much for your reply. We are glad that the explanation above has clarified things. We will make sure to include such discussions and incorporate your suggestions in our revision.

---

### Official Review · Reviewer_TbeN · 2022-07-12

**Rating:** 6
**Confidence:** 4
**Soundness:** 3 good
**Presentation:** 3 good
**Contribution:** 3 good

**Summary:**

This paper provides theoretical guarantees on multi-step off-policy distributional RL algorithms and derives a DRL algorithm, QR-DQN-Retrace based on QR-DQN which shows empirical improvements over QR-DQN on Atari.

**Questions:**

1. Is it possible to demonstrate the difference between multi-step value based algorithms and multi-step DRL algorithms in an empirical way?
2. As Figure 4 shows, the improvement of "Retrace" over "Uncorrelated" seems to be not significant.
3. How about applying the retrace idea on other DRL algorithms such as IQN, FQF.
4. Can this method ensures the validity of the learned distributions, such as the crossing issues?
Zhou, Fan, Jianing Wang, and Xingdong Feng. "Non-crossing quantile regression for distributional reinforcement learning." Advances in Neural Information Processing Systems 33 (2020): 15909-15919.

**Limitations:**

Although there are some theoretical contributions, the algorithm design is not novel enough, and also the empirical part needs to be improved to support the idea of the paper.

**Strengths And Weaknesses:**

There are two main contributions of this paper:
1. This paper systematically discusses the similarity and difference between multi-step value-based RL and multi-step distributional RL. They also provide theoretical guarantees on multi-step off-policy distributional RL algorithms.
2. This paper derives a novel algorithm, Quantile Regression-Retrace, which leads to a deep RL agent QR-DQN-Retrace that shows empirical improvements over QR-DQN on the Atari-57 benchmark.

There are some weaknesses of this paper.
1. Although there are some theoretical contributions, the methodological novelty is not that big from the perspective of algorithm architecture compared to the baseline QR-DQN.
2. The empirical part is not that solid while some more comparisons between multi-step value based algorithms and multi-step DRL algorithms are helpful.

---

> ### Author Response · Authors · 2022-08-02
> **Reply to your reviews**
>
> Thank you very much for your valuable inputs to our paper. We address your detailed comments below.
>
> ### **Question 1 on “difference between multi-step value-based and DRL algorithms"**
>
> While both multi-step learning and distributional RL improve value-based RL algorithms, their mechanisms of improvements are quite orthogonal to each other. In this work, since our focus is to apply principled multi-step learning techniques to distributional RL, we find it a bit distracting to add a comparison between multi-step value-based and DRL algorithms.
>
> Nevertheless, we can offer some comparison using results from prior literature. From the Rainbow paper [1], in Fig. 1, a comparable multi-step value-based algorithm achieves median performance at around 120%. In our work, multi-step distributional RL algorithms can generally achieve median performance at around 140%.
>
> [1] Hessel et al., Rainbow: combining improvements in deep RL. AAAI 2018.
>
> ### **Question 2 on “improvement of Retrace over uncorrected seems to be not significant”**
>
> In the median performance of C51 and mean performance of QR-DQN (Fig 4), distributional Retrace still obtains some statistically significant improvements over the uncorrected operator. This hints at a favorable effect of carrying out off-policy corrections with such algorithms. Note that such improvements are not as visible in the value-based setting from prior literature.
>
> In addition, in **Appendix I**, we carry out ablation studies that compare distributional Retrace and uncorrected n-step. In situations where off-policy corrections are essential to performance (such as when the bootstrapping horizon n is large, or when the data is very off-policy), distributional Retrace is clearly more robust and achieves significant improvements over uncorrected n-step methods.
>
> ### **Question 3 on “applying the idea to IQN”**
>
> We focus on C51 and QR-DQN, mainly because these two algorithms directly stem from theoretically-grounded tabular learning algorithms such as categorical TD learning and quantile TD learning. Since IQN and FQF are empirical extensions of C51 and QR-DQN, we leave such further empirical evaluations to future work.
>
> ### **Question 4 on “crossing quantile issues”**
>
> The multi-step method we introduced does not address the crossing issues out-of-the-box. However, non-crossing quantile regression could certainly be combined with our approach.

---

### Author Response · Authors · 2022-08-02
**Reply to all reviewers**

Thank you very much for all reviewers’ valuable inputs to our paper.

### **Improvements of Retrace over uncorrected n-step**

A few reviewers mention that distributional Retrace does not appear to significantly improve over uncorrected n-step in the deep RL experiments. We want to mention that the strong performance of uncorrected n-step works is a well-established empirical observation in value-based learning in general (n=4 in [1]; n=5-10 in [2,3]; n=10 for Atari 100k in [4]), and is not restricted to distributional RL. This might be attributed to the fact that agents like C51 and QR-DQN uses a epsilon-greedy policy that closely tracks the greedy target policy (e.g., over time epsilon decreases), making the bias of uncorrected methods more benign when the bootstrapping horizon $n$ is small [8].

However, this should not undermine the importance of multi-step operators whose fixed point is unbiased. The improvements of Retrace over uncorrected n-step become much more significant when the lack of off-policy corrections has a greater impact on the update target. Such situations include when the multi-step algorithm uses a very long bootstrapping horizon (e.g., using a large $n$) or when the behavior policy does not closely track the target policy (e.g., offline RL [5] or tandem RL setting [6]).

In **Appendix I**, we carry out ablation studies on the above cases. The overall performance of uncorrected n-step clearly degrades as the bootstrapping horizon n increases, whereas distributional Retrace is robust to the choice of n. Distributional Retrace also obtains a significant advantage over uncorrected n-step when the behavior policy is uniform, in which case proper off-policy corrections clearly help improve performance.

### **Summary of main changes in the uploaded revisions**

We made a few major changes to the appendix:

**Appendix H** includes a more in-depth discussion on the critical differences between path-dependent and path-independent multi-step distributional TD errors and their operators. We provide a formal proof of the fact that multi-step operators defined by path-independent multi-step distributional TD errors are **non-contractive**. This is meant to supplement the numerical experiment in the main paper.

**Appendix I** includes additional comparison between distributional Retrace and uncorrected $n$-step. In cases where the lack of off-policy corrections is detrimental to performance, Retrace outperforms uncorrected $n$-step more significantly. This also implies that in general Retrace is much more robust to hyper-parameters such as bootstrap horizon $n$ and replay ratio [7].

**Appendix J** includes per-game scores and additional game score reports for the experiments in Fig 4.

### **Reference**

[1] Hessel et al., Rainbow: combining improvements for deep reinforcement learning, AAAI 2018.

[2] Kapturowski et al., Recurrent experience replay in distributed reinforcement learning, ICLR 2018.

[3] Rowland et al., Adaptive trade-off in off-policy learning, AISTATS 2020.

[4] Schwarzer et al., Data efficient reinforcement learning with self-predictive representations, NeurIPS 2020.

[5] Levine et al., Tutorial on offline reinforcement learning, 2019.

[6] Ostrovski et al., The difficulty of passive learning in deep reinforcement learning, NeurIPS 2021.

[7] Fedus et al., Revisiting Fundamentals of Experience Replay, ICML 2020.

[8] Kozuno et al., Revisiting Peng's Q($\lambda$) for modern reinforcement learning, ICML 2021.

---

### Meta-Review · Area_Chair_t8ko · 2022-08-25

**Recommendation:** Accept
**Confidence:** Less certain

**Metareview:**

The reviewers carefully analyzed this work and agreed that the topics investigated in this paper are important and relevant to the field. Although the reviewers generally expressed positive views on the proposed method, they also pointed out many possible limitations of this paper. On the one hand, one reviewer acknowledged that the authors provided the first theoretical guarantees on multi-step off-policy distributional RL algorithms via a novel algorithm. They argued, however, that the methodological novelty may not be too significant compared to the baseline QR-DQN, and that the experiments could be improved. After reading the authors' rebuttal, this reviewer—although still with an overall positive impression of the quality of this work—said that they do not believe the paper fully shows the particularity of applying the multi-step idea to DRL when compared to some value-based methods. They also believe that the empirical section of the paper was not strong enough. Another reviewer agreed that this is novel and significant work, as it introduces a non-trivial extension of one-step distributional RL to the multi-step off-policy setting. They argued that this paper provides a theoretical basis for the continued study of multi-step paradigms in off-policy reinforcement learning, which is important/significant. This reviewer, however, pointed out that the experimental results were not sufficiently convincing to support the claims that employing the multi-step distributional RL algorithm provides a significant empirical advantage. This reviewer carefully analyzed the authors' rebuttal and appreciated that they clarified the reviewer's original points of confusion. Overall, however, this reviewer agreed with others that the empirical results are not overly convincing. Furthermore, one of the reviewers argued that the notion of path-dependent distributional TD error is novel. They pointed out, as the main limitation of this paper, that investigating the multi-step distributional RL setting "may not be [particularly interesting since] some theoretical results on one-step distributional RL can be straightforward to extend into the multi-step version". They also argued that the unbiased QR-loss was not clearly motivated since the bias for RL may also be useful for exploration. After reading the authors' rebuttal, this particular reviewer updated their score since the authors provided more evidence to strengthen their theoretical and empirical claims. Overall, all reviewers were positively impressed with the quality of this work but brought up many points of contention regarding ways in which the paper could/should still be improved. They encourage the authors to update their work based on their constructive criticisms and, in particular, in a way that tackles the points of contention mentioned in the original reviews and their post-rebuttal comments.

**Award:**

No

---

### Decision · Program_Chairs · 2022-09-14

Accept